# Patient-specific mutations impair BESTROPHIN1's essential role in mediating Ca$^{2+}$-dependent Cl$^-$ currents in human RPE

Yao Li[1†], Yu Zhang[2†], Yu Xu[1,3], Alec Kittredge[2], Nancy Ward[2], Shoudeng Chen[4], Stephen H Tsang[1]*, Tingting Yang[2]*

[1]Jonas Children's Vision Care, and Bernard and Shirlee Brown Glaucoma Laboratory, Department of Ophthalmology and Pathology & Cell Biology, Edward S. Harkness Eye Institute, New York Presbyterian Hospital/Columbia University, New York, United States; [2]Department of Pharmacology and Physiology, School of Medicine and Dentistry, University of Rochester, Rochester, United States; [3]Department of Ophthalmology, Xinhua Hospital affiliated to Shanghai Jiao Tong University School of Medicine, Shanghai, China; [4]Molecular Imaging Center, Department of Experimental Medicine, The Fifth Affiliated Hospital of Sun Yat-sen University, Zhuhai, China

*For correspondence:
sht2@cumc.columbia.edu (SHT);
tingting_yang@urmc.rochester.
edu (TY)

[†]These authors contributed
equally to this work

Competing interests: The
authors declare that no
competing interests exist.

Reviewing editor: Jeremy
Nathans, Johns Hopkins
University School of Medicine,
United States

**Abstract** Mutations in the human *BEST1* gene lead to retinal degenerative diseases displaying progressive vision loss and even blindness. BESTROPHIN1, encoded by *BEST1*, is predominantly expressed in retinal pigment epithelium (RPE), but its physiological role has been a mystery for the last two decades. Using a patient-specific iPSC-based disease model and interdisciplinary approaches, we comprehensively analyzed two distinct *BEST1* patient mutations, and discovered mechanistic correlations between patient clinical phenotypes, electrophysiology in their RPEs, and the structure and function of BESTROPHIN1 mutant channels. Our results revealed that the disease-causing mechanism of *BEST1* mutations is centered on the indispensable role of BESTROPHIN1 in mediating the long speculated Ca$^{2+}$-dependent Cl$^-$ current in RPE, and demonstrate that the pathological potential of *BEST1* mutations can be evaluated and predicted with our iPSC-based 'disease-in-a-dish' approach. Moreover, we demonstrated that patient RPE is rescuable with viral gene supplementation, providing a proof-of-concept for curing *BEST1*-associated diseases.

DOI: https://doi.org/10.7554/eLife.29914.001

## Introduction

The human *BEST1* gene encodes a protein (BESTROPHIN1, or BEST1) that is predominantly expressed in retinal pigment epithelium (RPE) (*Marmorstein et al., 2000*; *Marquardt et al., 1998*; *Petrukhin et al., 1998*). To date, over 200 distinct mutations in *BEST1* have been identified to associate with bestrophinopathies (http://www-huge.uni-regensburg.de/BEST1_database/home.php?select_db=BEST1), which consist of at least five distinct retinal degeneration disorders, namely: Best vitelliform macular dystrophy (BVMD) (*Marquardt et al., 1998*; *Petrukhin et al., 1998*), autosomal recessive bestrophinopathy (ARB) (*Burgess et al., 2008*), adult-onset vitelliform dystrophy (AVMD) (*Allikmets et al., 1999*; *Krämer et al., 2000*), autosomal dominant vitreoretinochoroidopathy (ADVIRC) (*Yardley et al., 2004*), and retinitis pigmentosa (RP) (*Davidson et al., 2009*). Patients with bestrophinopathies are susceptible to progressive vision loss for which there is currently no

**eLife digest** Mutations to the gene that encodes a protein called BESTROPHIN1 cause a number of human diseases that lead to a progressive loss of sight and even blindness. Over two hundred of these disease-causing mutations exist, but it is not understood how they affect BESTROPHIN1. Furthermore, there are currently no treatments available to treat these diseases.

BESTROPHIN1 is an ion channel found in cell membranes in the retinal pigment epithelium (RPE), a layer of cells in the eye that is vital for vision. When BESTROPHIN1 is stimulated by calcium ions, it opens up to allow chloride ions to flow into and out of the cell.

The health of human eyes can be assessed by measuring how well they respond to light – a response that is believed to be generated from the flow of calcium-stimulated chloride ions in the RPE. Patients with mutant BESTROPHIN1 channels have an abnormally low response to light, but it remains unclear whether these channels are responsible for maintaining the flow of chloride ions required for the light response. Indeed, it is not confirmed whether calcium-stimulated chloride flow occurs on the surface of normal human RPE cells at all.

Human RPE cells are difficult to obtain. Instead, Li, Zhang et al. took human skin cells – some from patients who had disease-causing mutations that affect BESTROPHIN1 – and used stem cell technology to coax the cells to develop into RPE cells. Calcium-stimulated chloride ion flow could be recorded on the surface of these cells.

Next, the impact of two disease-causing mutations on BESTROPHIN1 was examined. The mutation from the patient who displayed the more severe illness completely inactivated the channel, while the other associated with milder illness caused a partial loss of channel activity. Notably, introducing normal BESTROPHIN1 into the RPE cells developed from patients with mutant BESTRPOPHIN1 restored chloride ion flow to normal levels. Thus it appears that BESTROPHIN1 is essential for maintaining calcium-stimulated chloride ion flow in human RPE cells.

The techniques developed by Li, Zhang et al. form a patient-specific 'disease-in-a-dish' approach that could be used to study the consequences of other mutations to the gene that produces BESTROPHIN1. This work also suggests that gene therapy could potentially help to treat BESTROPHIN1-related diseases.

DOI: https://doi.org/10.7554/eLife.29914.002

treatment available. Therefore, understanding how disease-causing mutations affect the biological function of BEST1 in the retina is critical for elucidating the pathology of bestrophinopathies and developing rational therapeutic interventions.

A clinical feature of bestrophinopathies associated with *BEST1* mutations is abnormal electrooculogram (EOG) light peak (LP), measured by the maximum transepithelial potential produced by RPE upon light exposure (*Boon et al., 2009*; *Marmorstein et al., 2009*). LP is believed to represent a depolarization of the basolateral membrane of RPE due to activation of a $Cl^-$ conductance triggered by changes in intracellular $Ca^{2+}$ concentration ($[Ca^{2+}]_i$) (*Fujii et al., 1992*; *Gallemore and Steinberg, 1989*; *Gallemore and Steinberg, 1993*). The simplest hypothesis about the origin of this ion conductance is that it is generated by $Ca^{2+}$-activated $Cl^-$ channels (CaCCs). However, the existence of $Ca^{2+}$-dependent $Cl^-$ current on the plasma membrane of RPE has not yet been directly demonstrated, let alone the identity of the participating channel(s).

BEST1 localizes to the basolateral membrane of RPE (*Marmorstein et al., 2000*), and has been functionally identified as a CaCC in heterologous expression studies (*Hartzell et al., 2008*; *Kane Dickson et al., 2014*; *Sun et al., 2002*; *Tsunenari et al., 2003*; *Xiao et al., 2008*; *Yang et al., 2014b*). Consequently, whether or not BEST1 conducts $Ca^{2+}$-dependent $Cl^-$ currents responsible for LP in RPE has been a long-standing question in the field (*Hartzell et al., 2008*; *Johnson et al., 2017*). *Best1* knock-out mice do not have any retinal phenotype or $Cl^-$ current abnormality (*Marmorstein et al., 2006*; *Milenkovic et al., 2015*), suggesting that either BEST1 is not the $Cl^-$ conducting channel in RPE, or that there are fundamental differences between humans and mice regarding the genetic bases for this electrophysiological response. So far only two studies investigated the $Cl^-$ channel function of endogenous BEST1 in human RPE. Although both studies demonstrated that $Cl^-$ secretions were partially impaired in iPSC-RPEs (RPE cells differentiated from induced pluripotent

stem cells) derived from *BEST1* patients compared to those from healthy donors (*Milenkovic et al., 2015*; *Moshfegh et al., 2016*), whether or not the CaCC function of BEST1 is involved remains unknown. The first study measured volume-regulated Cl⁻ current without testing the involvement of $Ca^{2+}$, and used only one WT iPSC-RPE as the control which may not be representative (*Johnson et al., 2017*; *Milenkovic et al., 2015*). The second study, by our group, utilized anion sensitve fluorescent dyes to detect changes in $Ca^{2+}$-stimulated Cl⁻ secretion, which is not a direct measurement of CaCC activity (*Moshfegh et al., 2016*). As BEST1 has also been suggested to regulate intracellular $Ca^{2+}$ homeostasis by controlling intracellular $Ca^{2+}$ stores on the endoplasmic reticulum (ER) membrane and/or modulating $Ca^{2+}$ entry through L-type $Ca^{2+}$ channels (*Barro-Soria et al., 2010*; *Constable, 2014*; *Gómez et al., 2013*; *Neussert et al., 2010*; *Singh et al., 2013*; *Strauß et al., 2014*), our observations could potentially reflect BEST1's role as a regulator of $Ca^{2+}$ signaling rather than as a CaCC. Moreover, two recent reports argued that other CaCCs rather than BEST1 are responsible for $Ca^{2+}$-stimulated Cl⁻ current based on results from porcine and mouse RPEs, and the human RPE-derived ARPE-19 cell line (*Keckeis et al., 2017*; *Schreiber and Kunzelmann, 2016*).

Overall, the physiological role of BEST1 in human RPE and the pathological mechanisms of *BEST1* disease-causing mutations are still poorly understood. Here for the first time, we directly measured $Ca^{2+}$-dependent Cl⁻ currents on the plasma membrane of human RPEs by whole-cell patch clamp, evaluated the physiological influence of two distinct ARB patient-derived *BEST1* mutations in this context, and demonstrated rescue of mutation-caused loss of function by complementation. We

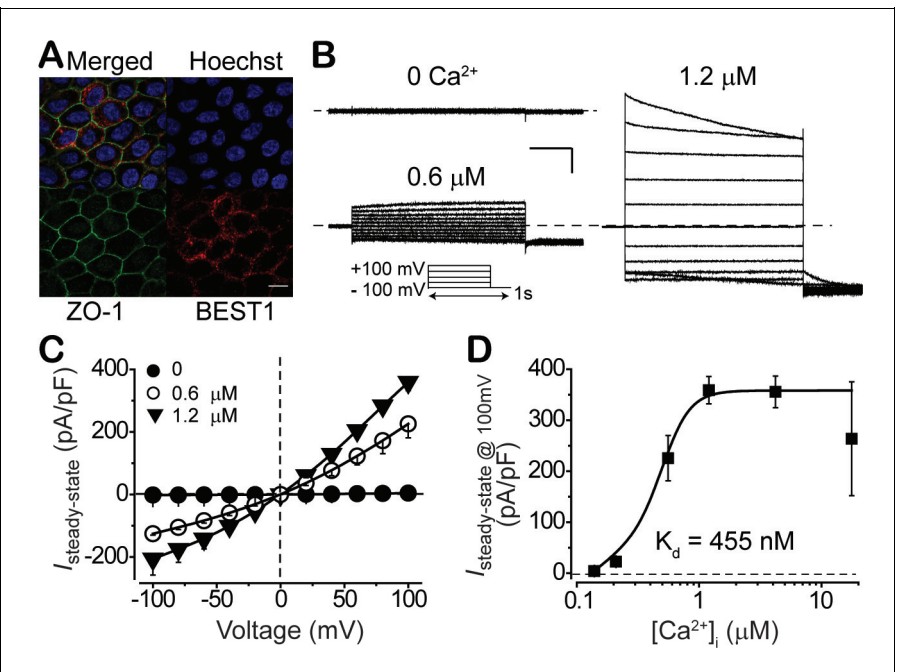

**Figure 1.** Subcellular localization of BEST1 and surface $Ca^{2+}$-dependent Cl⁻ current in *BEST1* WT donor iPSC-RPEs. (**A**) Confocal images showing plasma membrane localization of BEST1. Scale bar, 10 μm. (**B**) Representative current traces recorded from a *BEST1* WT donor iPSC-RPEs at various free $[Ca^{2+}]_i$. Voltage protocol used to elicit currents is shown in *Insert*. Scale bar, 1 nA, 150 ms. (**C**) Population steady-state current-voltage relationships at different free $[Ca^{2+}]_i$; n = 5–6 for each point. The plot was fitted to the Hill equation. (**D**) $Ca^{2+}$-dependent activation of surface current. Steady-state current density recorded at +100 mV plotted vs. free $[Ca^{2+}]_i$; n = 5–6 for each point. See also *Figure 1—figure supplements 1* and *Figure 1—source data 1*.
DOI: https://doi.org/10.7554/eLife.29914.003

The following source data and figure supplement are available for figure 1:

**Source data 1.** Comparison of different data sets from the same donors.
DOI: https://doi.org/10.7554/eLife.29914.005

**Figure supplement 1.** Characterization of WT iPSC and iPSC-RPE.
DOI: https://doi.org/10.7554/eLife.29914.004

further investigated the impacts of the two disease-causing mutations on the function and structure of BEST1 by electrophysiological and crystallographic approaches, respectively, and discovered mechanistic bases correlated with patient clinical phenotypes. Our results provide definitive evidence that the CaCC activity of BEST1 is indispensable for $Ca^{2+}$-dependent $Cl^-$ currents in human RPE, reveal the molecular mechanisms of two *BEST1* patient mutations, and offer a proof-of-concept for curing *BEST1*-associated retinal degenerative diseases.

## Results

### Direct recording of $Ca^{2+}$-dependent $Cl^-$ current by whole-cell patch clamp in human RPEs

Reduced LP is a pathognomonic phenotype associated with *BEST1* mutations in bestrophinopathy patients (*Boon et al., 2009*; *Marmorstein et al., 2009*). Although LP is believed to be mediated by surface $Ca^{2+}$-dependent $Cl^-$ current in RPE, the existence of the current on the plasma membrane of RPE cells has not been directly demonstrated, let alone the putative physiological role of BEST1 as a contributor to the current. To address these deficits, we generated iPSC-RPEs from the skin fibroblasts of two *BEST1* WT donors (*Figure 1—figure supplement 1A,B*). We first examined the subcellular localization of BEST1 by fluorescent co-immunostaining of the channel together with a plasma membrane marker (zonula occludens-1, ZO-1) and a nucleus marker (Hoechst) followed by confocal microscopy. We found that BEST1 localized on the plasma membrane of iPSC-RPE (*Figure 1A*, and *Figure 1—figure supplement 1C*).

We examined the $Ca^{2+}$-dependent $Cl^-$ current amplitudes on the plasma membrane of RPE using whole-cell patch clamp across a range of free $[Ca^{2+}]_i$ (*Figure 1B–D*, and *Figure 1—figure*

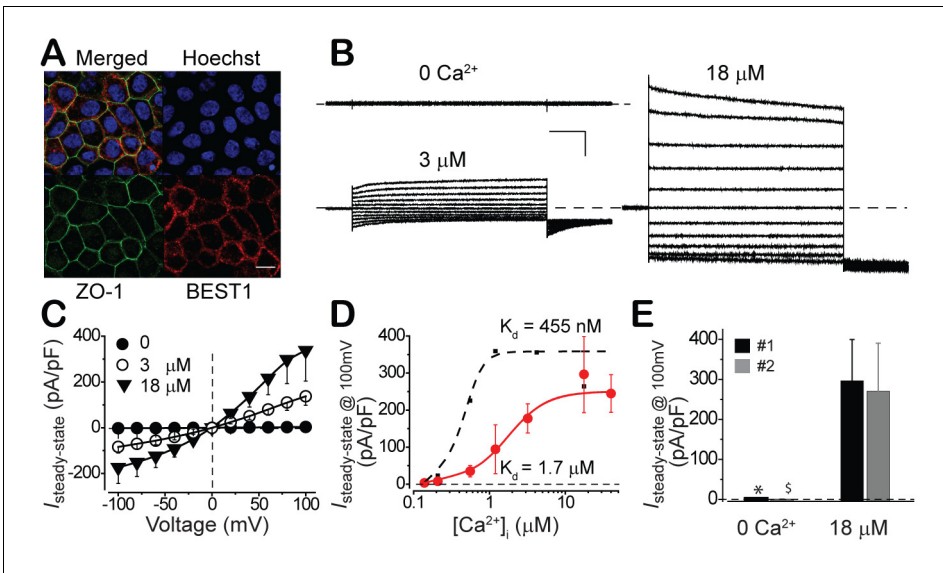

**Figure 2.** Subcellular localization of BEST1 and surface $Ca^{2+}$-dependent $Cl^-$ current in fhRPEs. (A) Confocal images showing plasma membrane localization of BEST1. Scale bar, 10 μm. (B) Representative current traces recorded from a *BEST1* WT fhRPEs at various free $[Ca^{2+}]_i$. Scale bar, 1 nA, 150 ms. (C) Population steady-state current-voltage relationships at different free $[Ca^{2+}]_i$; n = 5–6 for each point. (D) $Ca^{2+}$-dependent activation of surface currents in fhRPE (•) and iPSC-RPE (•). Steady-state current density recorded at +100 mV plotted vs. free $[Ca^{2+}]_i$; n = 5–6 for each point. The plots were fitted to the Hill equation. (E) Bar chart showing the steady-state current amplitudes at 0 and 18 μM free $[Ca^{2+}]_i$ in RPEs from two distinct human fetuses; n = 5–6. $^{*\$}p<0.05$ compared to fetus #1 (0.02) and #2 (0.02), respectively, at 18 μM $[Ca^{2+}]_i$ using two-tailed unpaired Student t test. See also *Figure 2—figure supplement 1*.

DOI: https://doi.org/10.7554/eLife.29914.006

The following figure supplement is available for figure 2:

**Figure supplement 1.** The $Ca^{2+}$ and time-dependent activation of surface $Cl^-$ current in fhRPE.

DOI: https://doi.org/10.7554/eLife.29914.007

supplement 1D). Currents were tiny (< 5 pA/pF) when $[Ca^{2+}]_i$ was 0 (*Figure 1B,C*), and increased in amplitude as $[Ca^{2+}]_i$ was raised from 100 nM to 4.2 μM, peaking at 358 ± 15 pA/pF at 1.2 and 4.2 μM $[Ca^{2+}]_i$ (*Figure 1B–D*, *Figure 1—figure supplement 1D*, and *Figure 1—source data 1*). The measured currents were inhibited by the Cl⁻ channel blocker niflumic acid (NFA) (*Figure 1—figure supplement 1D*), demonstrating that these were indeed $Ca^{2+}$-dependent Cl⁻ currents. A plot of peak current (evoked with a +100 mV step pulse) as a function of $[Ca^{2+}]_i$ displayed robust $Ca^{2+}$-dependent activation with the half maximal effective concentration ($EC_{50}$) of $Ca^{2+}$ at 455 nM. Similar $Ca^{2+}$-dependent Cl⁻ current profiles were recorded in iPSC-RPEs derived from two independent *BEST1* WT donors, and in iPSC-RPEs from two distinct clonal iPSCs of the same donor (*Figure 1—figure supplement 1*, and *Figure 1—source data 1*). These results provide the first direct measurement of $Ca^{2+}$-dependent Cl⁻ currents on the plasma membrane of RPE.

To test if the status of BEST1 and the properties of surface $Ca^{2+}$-dependent Cl⁻ current in iPSC-RPE represent those in real RPE, we conducted the same set of experiments in fetal human RPE (fhRPE). Consistent with the results from iPSC-RPEs, BEST1 was plasma membrane enriched (*Figure 2A*), and a similar pattern of $Ca^{2+}$-dependent Cl⁻ currents was recorded in fhRPEs from two independent fetuses (*Figure 2B–E*). Interestingly, despite their comparable initial and peak amplitudes, the $Ca^{2+}$-dependent Cl⁻ current in fhRPEs displayed a lower $Ca^{2+}$ sensitivity compared to that in iPSC-RPEs ($EC_{50}$ 1.7 μM vs. 455 nM, *Figure 2D*), which may reflect the different requirement of LP

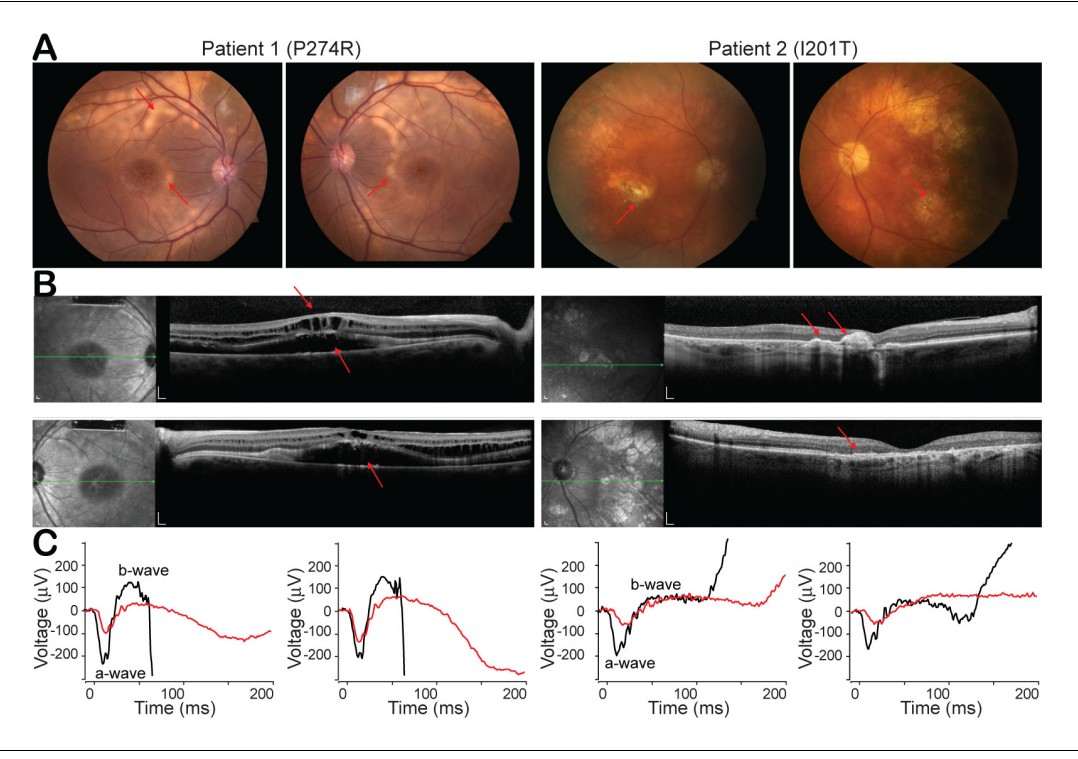

**Figure 3.** Clinical phenotypes of two patients with *BEST1* mutations. (**A**) Color fundus photographs from patient 1 (P274R) and patient 2 (I201T), right and left eyes, respectively. Both of the patients' fundus show bilateral, confluent curvilinear subretinal yellowish vitelliform deposits (red arrow) superior to the optic disks and encircling the maculae. (**B**) SDOCTs of the macula in patient 1 and patient 2. Scale bar, 200 μm. In Patient 1, there are bilateral, multifocal serous retinal detachments involving the maculae and cystoid deposits in the macula (red arrow). Patient 2 presents a relative preservation of the retina change compared to patient 1. (**C**) ERGs of patient 1 and patient 2 (red lines), right and left eyes, respectively, show extinguished maximum response amplitudes between a- and b-waves, compared to those from age matched *BEST1* WT controls (black lines). See also *Figure 3—figure supplement 1*.

DOI: https://doi.org/10.7554/eLife.29914.008

The following figure supplement is available for figure 3:

**Figure supplement 1.** Reduced EOG light peak in patient with *BEST1* I201T mutation.

DOI: https://doi.org/10.7554/eLife.29914.009

generation in RPE during different developmental stages. Overall, the subcellular localization of BEST1 and the properties of $Ca^{2+}$-dependent $Cl^-$ current in iPSC-RPE resemble those in fhRPE, validating iPSC-RPE as a powerful platform to study the influence of *BEST1* mutations on RPE surface $Ca^{2+}$-dependent $Cl^-$ currents.

It is worth to note that during patch clamp recording with fhRPE, when the pipet solution contained high (18 μM) $[Ca^{2+}]_i$, the currents ran up after patch break with a half-time of ~2.5 min and reached a plateau that was on average 7.8-fold greater than the initial current (*Figure 2—figure supplement 1A–C*). In contrast, when the pipet solution contained low (0.6 μM) $[Ca^{2+}]_i$, the currents remained stable after patch break (*Figure 2—figure supplement 1C*).

## Clinical phenotypes of two ARB patients with distinct *BEST1* mutations

Unlike the other bestrophinopathies caused by autosomal dominant mutations in *BEST1*, ARB is associated with recessive mutations. Patients with ARB are characterized by progressive generalized rod-cone degenerations, typically with a visual acuity reading around 20/40 in the first decade of life, and their vision progressively worsens over time (*Burgess et al., 2008*; *Johnson et al., 2017*). In this study, we focused on two diagnosed ARB patients from independent families. Both patients exhibit typical ARB phenotypes in fundus autofluorescence imaging, spectral domain optical coherence tomography (SDOCT) and full-field electroretinography (ERG) (*Figure 3A–C*). Unlike EOG which mainly represents the electrical responses of RPE (*Figure 3—figure supplement 1*), ERG measures the overall activity of various cell types in the retina.

Patient 1, a 12-year-old otherwise healthy boy, who has a previously described homozygous c.821C > G; p.P274R mutation in *BEST1* (*Fung et al., 2015*; *Kinnick et al., 2011*), showed reduced visual acuities at 20/60 and 20/70 in the right and left eye, respectively. Color fundus showed bilateral, confluent curvilinear subretinal yellowish vitelliform deposits to the optic disks, which over 3 years of follow-up became more multifocal and dispersed to involve the nasal retinae (*Figure 3A*, *left*). SDOCT discovered bilateral, multifocal serous retinal detachments involving the maculae and cystoid changes in the macula (*Figure 3B*, *left*). Maximum response of ERG b-wave (amplitudes between a- and b-wave) were 132.6 μV and 194.4 μV in the right and left eye, respectively, contrasting 355 μV (median value) in healthy teenagers tested in the same device (*Figure 3C*, *left*).

Patient 2, a 72-year-old otherwise healthy man, who has a homozygous c.602T > C; p.I201T mutation in *BEST1*, showed a dropped vision acuity at 20/40 in the right eye, and 20/200 in the left eye mainly due to aging-caused retinal atrophy. His color fundus presented less vitelliform deposits compared with patient 1, and aging-caused dispersed punctate fleck lesions in the left eye (*Figure 3A*, *right*). SDOCT showed milder cystic degeneration compared to that in Patient 1 (*Figure 3B*, *right*). Maximum responses of ERG b-waves were 103.2 μV and 79.6 μV in the right and left eye, respectively, contrasting 287 μV (median value) in age-matched healthy people (*Figure 3C*, *right*).

In summary, even though ARB has progressed for 60 years longer, patient 2 has better vision acuity (in his more relevant right eye), less vitelliform deposit, milder cystic degeneration, and better responses to visual stimuli, suggesting that the I201T mutation is less severe than the P274R mutation.

## Physiological impact of *BEST1* disease-causing mutations

If the recorded $Ca^{2+}$-dependent $Cl^-$ current is responsible for LP, it is logically speculated to be impaired in *BEST1* patient iPSC-RPEs, because reduced LP is a clinical feature in *BEST1* patients. To directly examine the physiological impact of *BEST1* mutations on $Ca^{2+}$-dependent $Cl^-$ current in RPE, iPSCs were derived from the patients' skin cells and then differentiated to iPSC-RPEs. RPE-specific marker proteins RPE65 (retinal pigment epithelium-specific 65 kDa protein) and CRALBP (cellular retinaldehyde-binding protein) displayed similar expression levels in the *BEST1* WT and two patient-derived iPSC-RPEs by western blot (*Figure 4A*), confirming the mature status of iPSC-RPEs.

Patient iPSC-RPE carrying the BEST1 P274R mutation showed a similar overall BEST1 expression level compared to that in WT iPSC-RPE (*Figure 4A*) in western blot, but exhibited diminished BEST1 antibody staining on the plasma membrane (*Figure 4B*, *top*), indicating that the subcellular localization of the channel was severely impaired by the P274R mutation. Strikingly, tiny currents (< 6 pA/pF) were detected in P274R patient iPSC-RPE at all tested $[Ca^{2+}]_i$ by whole-cell patch clamp (*Figure 4C*-E p, and *Figure 1—source data 1*), indicating that the P274R mutation abolishes $Ca^{2+}$-

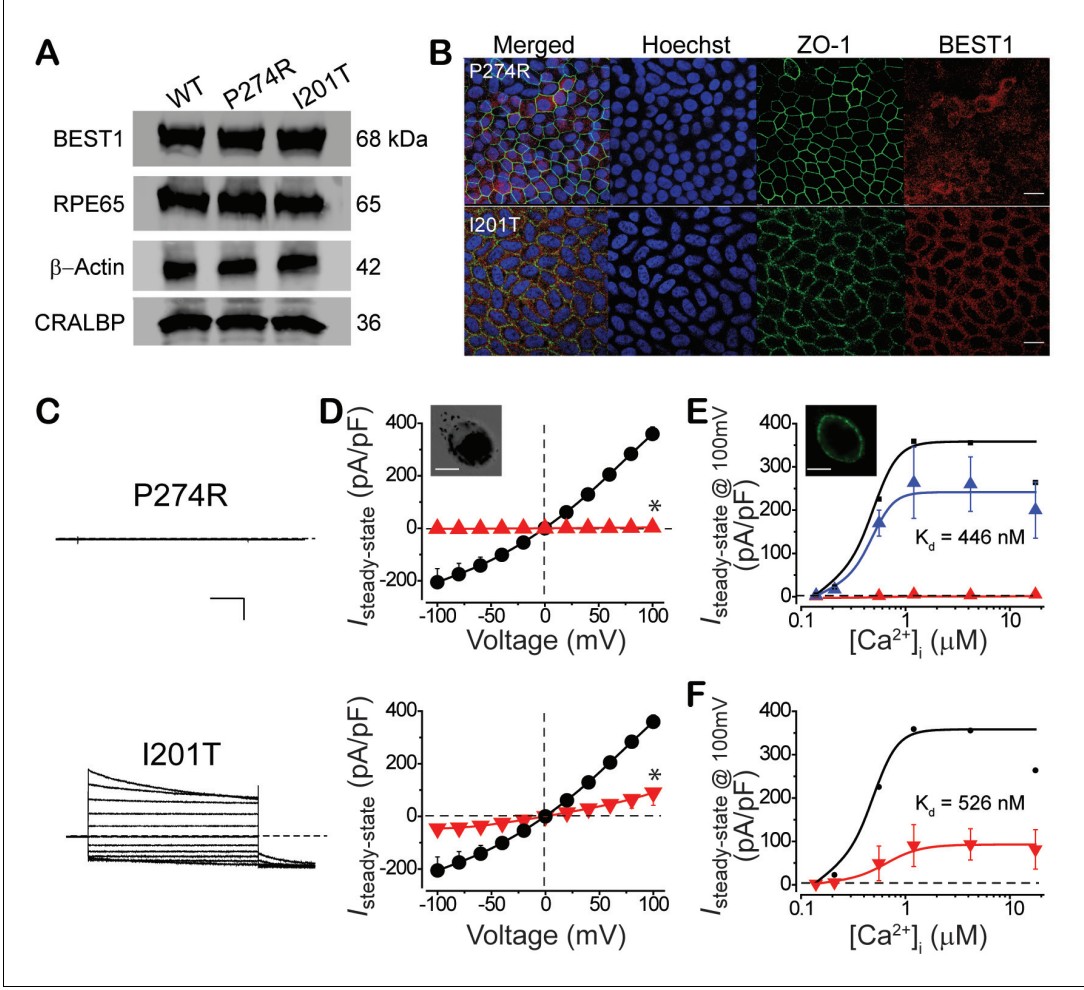

**Figure 4.** Subcellular localization of BEST1 and surface $Ca^{2+}$-dependent $Cl^-$ current in patient-derived iPSC-RPEs. (A) Western blots show similar BEST1 expression levels in WT and patient-derived iPSC-RPEs. Each sample was from one cell lysis (BEST1 and β-actin, RPE65 and CRALBP were on two gels, respectively). (B) Confocal images showing diminished plasma membrane localizations of BEST1 P274R, and normal plasma membrane localization of BEST1 I201T. Scale bar,15 μm. (C) Representative current traces recorded from patient iPSC-RPEs at 1.2 μM $[Ca^{2+}]_i$. Scale bar, 500 pA, 150 ms. (D) Population steady-state current-voltage relationships in BEST1 WT (•), P274R (▲) and I201T (▼) iPSC-RPEs at 1.2 μM $[Ca^{2+}]_i$; n = 5–6 for each point. *$p<0.05$ ($2 \times 10^{-7}$ for P274R and $6 \times 10^{-4}$ for I201T) compared to WT using two-tailed unpaired Student $t$ test. *Insert*, confocal images showing P274R iPSC-RPE in bright field. Scale bar,10 μm. (E) CaCC currents in BEST1 P274R patient iPSC-RPE were rescued by complementation with WT BEST1-GFP. Complementation (▲, n = 5–6 for each point), compared to BEST1 P274R (▲, n = 3–5 for each point), and WT (•). The plots were fitted to the Hill equation. *Insert*, confocal images showing P274R iPSC-RPE complemented with WT BEST1-GFP expressed from a BacMam baculoviral vector. Scale bar,10 μm. (F) $Ca^{2+}$-dependent currents in BEST1 I201T iPSC-RPE (▼) compared to WT iPSC-RPE (•). Steady-state current density recorded at +100 mV plotted vs. free $[Ca^{2+}]_i$; n = 5–6 for each point. The plots were fitted to the Hill equation. See also *Figure 4—figure supplement 1* and *Figure 1—source data 1*.
DOI: https://doi.org/10.7554/eLife.29914.010

The following figure supplement is available for figure 4:

**Figure supplement 1.** CaCC currents in *BEST1* patient iPSC-RPEs.
DOI: https://doi.org/10.7554/eLife.29914.011

dependent $Cl^-$ current in RPE. Furthermore, both the membrane localization of BEST1 and the $Ca^{2+}$-dependent $Cl^-$ current were rescued in P274R patient iPSC-RPE by complementation with WT BEST1-GFP expressed from a BacMam baculoviral vector (*Figure 4E*, and *Figure 4—figure supplement 1A,B,C*). These results demonstrated that functional BEST1 is necessary for generating $Ca^{2+}$-dependent $Cl^-$ current in human RPE.

On the other hand, patient iPSC-RPE carrying the BEST1 I201T mutation showed a similar overall BEST1 level compared to that in WT iPSC-RPE (*Figure 4A*), and normal BEST1 antibody staining on the plasma membrane (*Figure 4B*, *bottom*). However, I201T patient iPSC-RPE displayed robust but significantly decreased $Ca^{2+}$-dependent $Cl^-$ currents compared to those in WT iPSC-RPE (*Figure 4C, D,F*, and *Figure 1—source data 1*). Notably, when current amplitudes were normalized, the pattern of $Ca^{2+}$ response was similar in WT and I201T iPSC-RPEs ($EC_{50}$ 455 nM vs. 526 nM, *Figure 4F*, and *Figure 4—figure supplement 1D*), indicating that the $Ca^{2+}$ sensitivity of surface $Cl^-$ current in RPE is not altered by the I201T mutation.

Taken together, our results showed that the P274R mutation leads to a 'null' phenotype of $Ca^{2+}$-dependent $Cl^-$ current in RPE associated with a loss of BEST1 plasma membrane enrichment, while the seemingly milder I201T mutation causes reduced $Cl^-$ current in RPE without altering $Ca^{2+}$ sensitivity of the current or subcellular localization of BEST1. Importantly, the P274R patient exhibits a more severe retinal phenotype compared to the I201T patient, suggesting a strong correlation between the status of BEST1, the functionality of RPE surface $Ca^{2+}$-dependent $Cl^-$ current, and retinal physiology.

As BEST1 is a CaCC located on the plasma membrane of RPE, the next important question is whether the defective $Ca^{2+}$-dependent $Cl^-$ current in *BEST1* patient iPSC-RPEs truly reflects deficiency of the BEST1 channel activity. To directly examine the influence of the disease-causing mutations on BEST1, WT and mutant BEST1 channels were individually introduced into HEK293 cells, which do not have any endogenous CaCC on the plasma membrane (*Figure 5—figure supplement 1A,B*). Western blot confirmed that both WT and the mutant channels were expressed at similar levels after transient transfection (*Figure 5—figure supplement 1C*). As previously reported, HEK293 cells expressing WT BEST1 displayed robust $Ca^{2+}$-dependent currents markedly inhibited by NFA (*Figure 5—figure supplement 1B*), indicating that they were $Ca^{2+}$-dependent $Cl^-$ currents (*Hartzell et al., 2008*). Consistent with the results in iPSC-RPE, HEK293 cells expressing the P274R mutant yielded no current, while cells expressing the I201T mutant displayed significantly smaller current amplitude compared to that of WT at 1.2 µM $[Ca^{2+}]_i$, where HEK293 cells expressing WT BEST1 conduct peak current amplitude (*Figure 5A,B*) (*Hartzell et al., 2008*). As HEK293 cells represent a 'blank' background, the recorded $Ca^{2+}$-dependent $Cl^-$ currents are genuinely generated from transiently transfected BEST1 channels. Therefore, the two disease-causing mutations lead to distinct defects of the BEST1 channel activity that match the defects of $Ca^{2+}$-dependent $Cl^-$ current in iPSC-RPEs, strongly suggesting that BEST1 is the bona fide CaCC on the plasma membrane of RPE mediating $Ca^{2+}$-dependent $Cl^-$ current for LP.

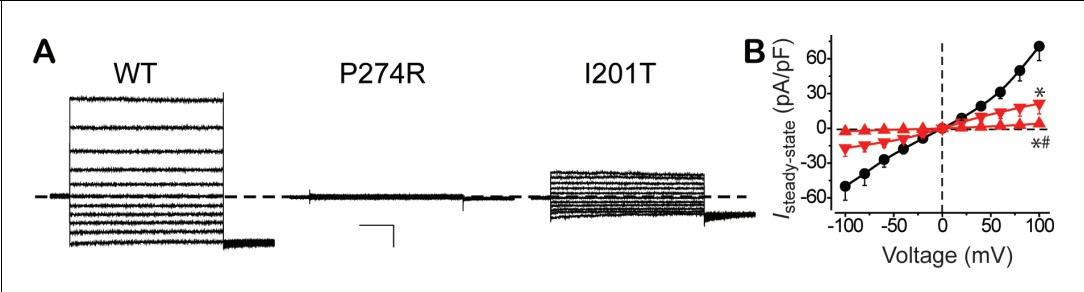

**Figure 5.** Surface $Ca^{2+}$-dependent $Cl^-$ current in HEK293 cells expressing WT and mutant BEST1. (**A**) Representative current traces recorded from transfected HEK293 cells at 1.2 µM $[Ca^{2+}]_i$. Scale bar, 150 pA, 150 ms. (**B**) Population steady-state current-voltage relationships for BEST1 WT (•), P274R (▲) and I201T (▼) at 1.2 µM $[Ca^{2+}]_i$; n = 5–6 for each point. *#$p<0.05$ compared to WT ($8 \times 10^{-4}$ for P274R and 0.01 for I201T) or to I201T (0.04), respectively, using one-way ANOVA and Bonferroni *post hoc* analyses. See also *Figure 5—figure supplement 1*.
DOI: https://doi.org/10.7554/eLife.29914.012

The following figure supplement is available for figure 5:

**Figure supplement 1.** $Ca^{2+}$-dependent $Cl^-$ current in BEST1 transfected HEK293 cells.
DOI: https://doi.org/10.7554/eLife.29914.013

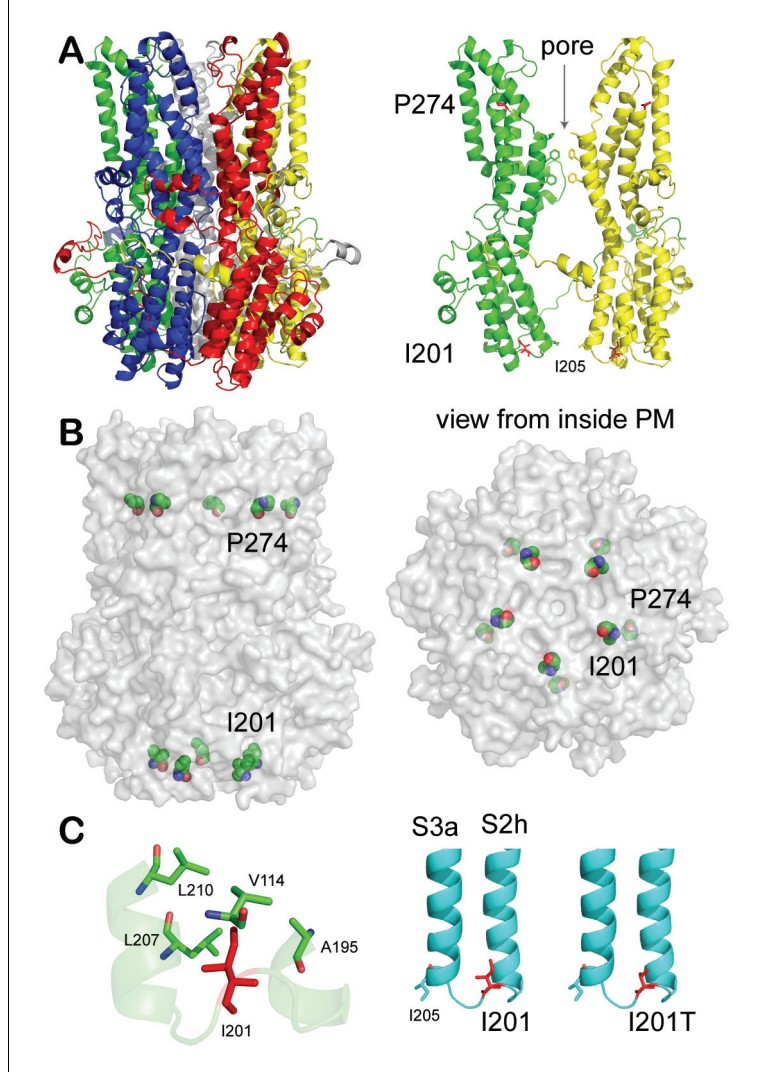

**Figure 6.** Patient mutations in a BEST1 homology model. (**A**) *Left*, ribbon diagram of the BEST1 pentamer with each protomer colored differently, as viewed from the side. *Right*, ribbon diagram of two oppositely facing (144°) protomers of a BEST1 pentamer are shown with the extracellular side on the top. The side chains of critical residues are in red. (**B**) Location of the patient mutations in relationship to the channel pore. *Left*, as viewed from the side; *right*, from inside the plasma membrane. (**C**) Visualization of the location of I201T. The side chains of critical residues are in red. See also *Figure 6—figure supplements 1* and *2*.

DOI: https://doi.org/10.7554/eLife.29914.014

The following figure supplements are available for figure 6:

**Figure supplement 1.** Structural analysis of *BEST1* mutations in a homology model.
DOI: https://doi.org/10.7554/eLife.29914.015

**Figure supplement 2.** Structure-based sequence alignment of KpBest, hBest1 and cBest1.
DOI: https://doi.org/10.7554/eLife.29914.016

## Disease-causing mechanisms of *BEST1* mutations

As an ion channel, how could BEST1 go wrong with the disease-causing mutations? Multiple mechanisms may exist, including massive disruption of the channel structure, alterations in single channel activity, and dysregulation of the channel (e.g. expression). We sought to find critical clues from the channel structure to answer this question.

Since the structure of BEST1 has not been solved, we generated a three-dimensional human homology model based on our previously solved *Klebsiella pneumoniae* bestrophin (KpBest) structure and a chicken bestrophin1 (cBest1) structure (*Kane Dickson et al., 2014*; *Moshfegh et al.,*

*2016*; *Yang et al., 2014b*) (*Figure 6A*, *Figure 6—figure supplement 1A,B*, and *Figure 6—figure supplement 2*). In this BEST1 model, P274 locates at the N-terminal of helix S4a (*Figure 6A,B*, *Figure 6—figure supplement 1A,B*, and *Figure 6—figure supplement 2*). The presence of Pro in alpha helices normally promotes thermostability of the membrane protein (*Reiersen and Rees, 2001*). The restricted torsion angle for the N–Cα bond of Pro allows only a limited number of conformations and imposes stress on secondary structures in proteins. Substitution of Pro with Arg will release the restrictions and induce instability of local structure, predicting a dramatic disruption of the channel. It should be noted that a Pro to Arg mutation based on the structure model would result in a steric clash between this amino acid and helix S3b, thereby highlighting the major contribution of Pro in the structure (*Figure 6—figure supplement 1D*).

On the other hand, I201 resides in a loop between S2h and S3a (*Figure 6A,B*, *Figure 6—figure supplement 1A,B*, and *Figure 6—figure supplement 2*), surrounded by hydrophobic residues V114, A195, L207, and L210 (*Figure 6C*), which are conserved among species and thus probably important for the channel function (*Figure 6—figure supplement 1C*). As the Ile to Thr substitution changes a hydrophobic residue to a polar residue, which weakens the hydrophobic interactions, this mutation may change the channel property by altering the local interplays between spatially adjacent subunits, but will unlikely disrupt the channel structure as its localization on a loop renders flexibility. Importantly, the potential influence of the I201T mutation on the channel function is underlined by its proximity to I205 (*Figure 6A,C*), a putative activation/permeation gate and the narrowest exit along the ion conducting pathway (*Figure 6A,B*) (*Yang et al., 2014b*).

Sequence alignment reveals that BEST1 P274 is identical while I201 has a highly conservative substitution in KpBest (P239 and L177, respectively, *Figure 6—figure supplement 1C*, and *Figure 6—figure supplement 2*), prompting us to test the predictions from the BEST1 homology model with the corresponding KpBest mutants (P239R and L177T, respectively) expressed from *E. coli*. During protein purification, we noticed that the yield of pentameric KpBest P239R was significantly less compared to that of KpBest WT or L177T (*Figure 7A*), consistent with the prediction that P274R

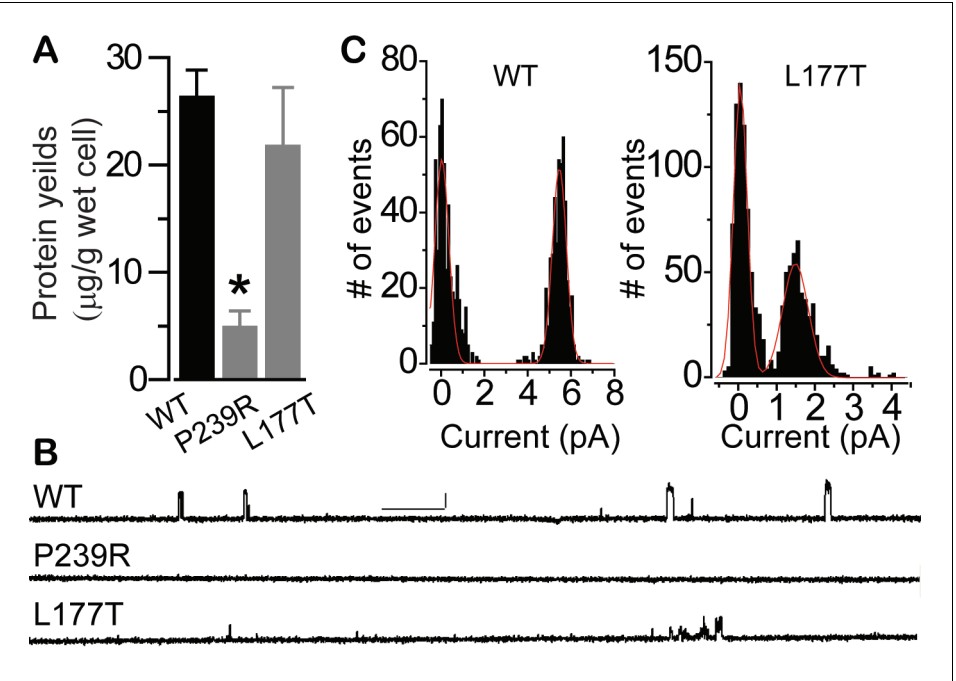

**Figure 7.** Influence of patient mutations on single channel conductance. (**A**) Bar chart showing purified KpBest WT and mutant pentameric protein per wet cell yields. n = 3 for each bar. *$p<0.05$ compared to WT ($2 \times 10^{-3}$) or L177T (0.03) using two-tailed unpaired Student $t$ test. (**B**) Current trace of KpBest WT and mutant single channels recorded from planar lipid bilayers at 80 mV with 150 mM NaCl in both cis and trans solutions. Scale bar, 2.5 pA, 250 ms. (**C**) Histograms showing single channel current amplitudes of KpBest WT and the L177T mutant. n = 3.
DOI: https://doi.org/10.7554/eLife.29914.017

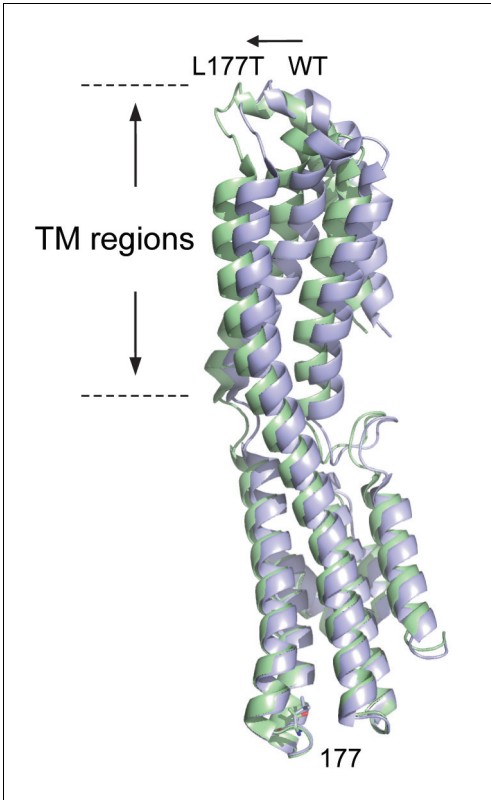

**Figure 8.** Superposition of KpBest WT with L177T mutant based on regional alignment of residues 174–180. Ribbon diagram of the KpBest WT chain A (blue) and KpBest L177T chain A (green) with highlighted stick diagram of residue 177. See also *Figure 8—figure supplement 1* and *Figure 8—source data 1*.
DOI: https://doi.org/10.7554/eLife.29914.018

The following source data and figure supplement are available for figure 8:

**Source data 1.** Data collection and refinement statistics of KpBest L177T.
DOI: https://doi.org/10.7554/eLife.29914.020

**Figure supplement 1.** Crystal structure of KpBest L177T.
DOI: https://doi.org/10.7554/eLife.29914.019

causes massive disruption, and thus instability, to the channel structure. Purified KpBest mutant proteins were set for crystal growing. While no crystal was grown with KpBest P239R, well-diffracted KpBest L177T crystals were obtained under the same condition as KpBest WT (*Yang et al., 2014b*), and the structure was solved to 3.1 Å resolution (*Figure 8—source data 1*). The KpBest L177T structure mirrors that of KpBest WT, with all-atom alignment RMSD (root-mean-square deviation) in a protomer 0.4 Å (Coot LSQ superpose). However, superposition of KpBest WT with the L177T mutant based on the alignment of single chain residues 174–180 showed an obvious shift of the TM region (*Figure 8*, and *Figure 8—figure supplement 1*). These results strongly support our structural predictions on the BEST1 P274R and I201T mutations.

We next assessed the influence of the disease-causing mutations on BEST1 single channel activity. To circumvent the unavailability of purified human BEST1, we utilized the corresponding KpBest P239R and L177T mutants. As previously described (*Yang et al., 2014b*), purified KpBest channels were fused into planar lipid bilayer with 150 mM NaCl in both the *trans* (internal) and *cis* (external) solutions, and single channel currents were recorded with KpBest WT at 80 mV with mean amplitude of 5.5 pA (*Figure 7B*). By contrast, no currents were obtained with KpBest P239R, while currents with reduced unitary conductance (mean amplitude 1.5 pA) were recorded with KpBest L177T (*Figure 7B,C*), suggesting that the BEST1 P274R and I201T mutations result in a complete and partial loss of single channel activity, respectively. Taken together, we concluded that P274R is a null mutation that abolishes both plasma membrane localization and channel activity of BEST1 due to structural disruption, whereas I201T is a partial loss-of-function mutation that retains plasma membrane localization and $Ca^{2+}$ sensitivity of BEST1 caused by minor structural alterations.

## Discussion

Here, we first proved the existence of $Ca^{2+}$-dependent $Cl^-$ currents on the plasma membrane of human RPE by whole-cell patch clamp. Then we comprehensively examined two *BEST1* disease-causing mutations (P274R and I201T) derived from ARB patients in an interdisciplinary platform, including whole-cell patch clamp with patient-derived iPSC-RPEs and HEK293 cells, immunodetection of endogenous BEST1 in iPSC-RPEs, lipid bilayer with purified bacterial bestrophin proteins, and structural analyses with human models and bacterial homolog crystal structures (*Table 1*). Collectively, our results illustrated the physiological influence of these two mutations on RPE surface $Ca^{2+}$-dependent $Cl^-$ current and the BEST1 channel function, and provided structural insights into their disease-causing mechanisms: the P274R mutation abolishes $Ca^{2+}$-dependent $Cl^-$ current in vivo, likely due to

**Table 1.** Summary of disease-causing mechanisms of BEST1 P274R and I201T mutations.

| | Mechanism | System | P274R | I201T |
|---|---|---|---|---|
| Phenotype | - | | Patient | Severe | Mild |
| Function | $I$ | CaC current | RPE | Null | Small |
| | | $Ca^{2+}$ sensitivity | RPE | N/A | Normal |
| | | CaC current of BEST1 | HEK293 | Null | Small |
| | $N$ | BEST1 expression | RPE | Normal | Normal |
| | | Membrane localization | RPE | Diminished | Normal |
| | $i$ | Unitary current | KpBest | Null | Small |
| Structure | - | | KpBest crystal + human model | Disrupted | Slightly altered |

$I = N \times P_o \times i$. $I$, whole-cell current amplitude; $N$, number of surface channels; $P_o$, channel open probability; $i$, unitary current.

DOI: https://doi.org/10.7554/eLife.29914.021

disruption of the BEST1 channel structure; while the I201T mutation partially impairs $Ca^{2+}$-dependent $Cl^-$ current in vivo, likely due to non-disruptive structural alteration (*Table 1*).

The structure of BEST1 has not been solved, and only two bestrophin homolog structures- KpBest and cBest1, were reported in previous studies (*Kane Dickson et al., 2014*; *Yang et al., 2014b*). We used both KpBest crystal structures and human homology models mainly based on cBest1 to analyze the possible structural alterations in BEST1 caused by the patient-specific mutations. Results from the two methods are consistent with each other and with functional data. Moreover, it has been proposed that disease mutations may result in wrongly numbered oligomers rather than the correct pentamer formed by WT BEST1 (*Johnson et al., 2017*). The structure of KpBest I177T suggests that the BEST1 I201T mutation does not alter the pentameric conformation of the channel.

Although decreased LP in *BEST1* patients has been attributed to aberrant RPE surface $Ca^{2+}$-dependent $Cl^-$ current, how *BEST1* disease-causing mutations physiologically influence $Ca^{2+}$-dependent $Cl^-$ current in RPE has not been directly examined. Most previous studies investigated the anion channel function of BEST1 in transiently transfected cell lines (*Hartzell et al., 2008*; *Johnson et al., 2017*), while the only two studies done in human RPE by other groups did not directly examine $Ca^{2+}$-dependent $Cl^-$ current: one measured transepithelial potential in fhRPE expressing exogenous BEST1 on virus (*Marmorstein et al., 2015*), and the other investigated volume-dependent current (*Milenkovic et al., 2015*). We recently used anion sensitive fluorescent dyes to compare $Ca^{2+}$-stimulated $Cl^-$ secretion in *BEST1* WT and mutant donor iPSC-RPEs, but neither the surface $Cl^-$ current nor $Ca^{2+}$ sensitivity was directly measured (*Moshfegh et al., 2016*). Here we clearly demonstrated with whole-cell patch clamp that the surface $Ca^{2+}$-dependent $Cl^-$ current in patient-derived iPSC-RPEs is completely abolished and significantly reduced by the P274R and I201T mutations, respectively, providing the first direct evidence that *BEST1* disease-causing mutations impair $Ca^{2+}$-dependent $Cl^-$ current in human RPE. Our results strongly argue that BEST1 is the CaCC mediating $Ca^{2+}$-dependent $Cl^-$ current in human RPE, because: 1) the surface $Ca^{2+}$-dependent $Cl^-$ current is completely defective in iPSC-RPE with the P274R mutation, which generates an essentially 'null' BEST1 channel with loss of plasma membrane enrichment in RPE and no ion conductivity in HEK293 cells and bilayer (KpBest P239R), suggesting that BEST1 is indispensable for $Ca^{2+}$-dependent $Cl^-$ current in RPE; 2) the I201T mutation results in significantly reduced conductivity of the channel in both HEK293 cells and bilayer (KpBest L177T), and concomitantly leads to much smaller $Ca^{2+}$-dependent $Cl^-$ currents in the patient iPSC-RPE, in which the mutant BEST1 channels are still expressed and enriched on the plasma membrane, suggesting that the CaCC function of membrane located BEST1 orchestrates $Ca^{2+}$-dependent $Cl^-$ current in RPE; 3) the I201T mutation does not affect the $Ca^{2+}$ sensitivity of $Cl^-$ current in RPE, consistent with the non-involvement of I201 in $Ca^{2+}$ binding according to the cBest1 crystal structure model (*Kane Dickson et al., 2014*). The simplest and most logical conclusion based on our results is that BEST1 functions as the surface CaCC to generate $Ca^{2+}$-dependent $Cl^-$ current in human RPE.

A recent report with primary mouse RPE and the human RPE-derived ARPE-19 cell line suggested that TMEM16B is the CaCC responsible for $Ca^{2+}$-stimulated $Cl^-$ current in those cells (*Keckeis et al., 2017*), while a role of TMEM16A was proposed by another study using $Cl^-$ channel blockers in

porcine RPE (*Schreiber and Kunzelmann, 2016*). It should be noted that *Best1* knockout mice do not have any retinal abnormality or aberrant Cl$^-$ current (*Marmorstein et al., 2006*; *Milenkovic et al., 2015*), unlike the phenotypes seen with human *BEST1* mutations, suggesting different genetic requirements for retinal physiology among species. Moreover, the expression of BEST1 in the ARPE-19 cell line may be different from that in iPSC-RPE and fhRPE (*Marmorstein et al., 2000*). In any case, our results do not completely exclude the role of other CaCCs in contributing to Ca$^{2+}$-dependent Cl$^-$ current in human RPE.

Besides its CaCC function on the basolateral plasma membrane of RPE, other roles of BEST1 have also been suggested including HCO$_3^-$ channel, volume-regulated Cl$^-$ channel, regulator of Ca$^{2+}$ channels, and Ca$^{2+}$ sensor on the endoplasmic reticulum membrane (*Barro-Soria et al., 2010*; *Burgess et al., 2008*; *Fischmeister and Hartzell, 2005*; *Gómez et al., 2013*; *Qu and Hartzell, 2008*; *Rosenthal et al., 2006*; *Yu et al., 2008*). Here we focused on the CaCC function of BEST1 in conducting surface Ca$^{2+}$-dependent Cl$^-$ current, which directly gives rise to LP, but did not exclude any indirect contribution of BEST1 to LP through its non-CaCC function(s). For instance, BEST1 may affect a downstream CaCC (e.g. TMEM16A or TMEM16B) through regulating intracellular Ca$^{2+}$.

Interestingly, we found remarkable differences in the Ca$^{2+}$ sensitivity of Cl$^-$ current in different cell types. The lower Ca$^{2+}$ sensitivities in fhRPE (EC$_{50}$ 1.7 μM) compared to that in *BEST1* WT iPSC-RPE (EC$_{50}$ 455 nM) may result from the cells' different developmental stages, considering that fetuses do not have a fully functional visual system and therefore probably only need less sensitive CaCCs in their RPEs. The higher Ca$^{2+}$ sensitivity of heterologously expressed BEST1 in HEK293 cells (EC$_{50}$ ~150 nM) has been reported in previous studies (*Lee et al., 2010*; *Xiao et al., 2008*), while purified cBest1 displays an even smaller EC$_{50}$ of 17 nM in bilayer (*Vaisey et al., 2016*). Considering the role of BEST1 as the CaCC in RPE, the significant difference of Ca$^{2+}$ sensitivities may reflect intrinsic differences between RPE where BEST1 is endogenously expressed and other experimental systems with overexpressed or purified proteins. It is also possible that in native RPE, BEST1 senses Ca$^{2+}$ not only through direct interaction as suggested by the cBest1 model (*Kane Dickson et al., 2014*), but also indirectly via a third-party Ca$^{2+}$-sensor protein or by posttranslational modification mechanisms (e.g. phosphorylation) (*Hartzell et al., 2008*), to function properly under the sophisticated physiological environment. Notably, the Ca$^{2+}$ sensitivity observed in *BEST1* WT iPSC-RPE (EC$_{50}$ 455 nM) is at levels more comparable to physiological conditions than that detected in HEK293 cells over-expressing BEST1 (EC$_{50}$ ~150 nM), let alone cBest1 in bilayer (EC$_{50}$ 17 nM), because basal [Ca$^{2+}$]$_i$ in the human body is typically around 100 nM, meaning that CaCCs with a EC$_{50}$ near or lower than 100 nM would be readily activated even in resting cells.

In regard to the clinical treatment of bestrophinopathies, our study provided an important proof-of-concept for treating ARB caused by *BEST1* recessive mutations, as the loss of Ca$^{2+}$-dependent Cl$^-$ current in the 'null' BEST1 P274R iPSC-RPE was rescued by viral expression of WT BEST1. It will be very intriguing to see if ARB patients can be treated by gene therapy delivering functional WT BEST1 to their RPEs. Notably, most of the *BEST1* patient-specific mutations are dominant, so that the mutant *BEST1* alleles in these cases may be more functionally defective and/or structurally disruptive compared to recessive mutant alleles in ARB patients. Although it is formally possible that overexpression of WT BEST1 can also rescue, in a dominant-negative matter, aberrant Ca$^{2+}$-dependent Cl$^-$ current in RPE caused by *BEST1* dominant mutations, further studies will be needed to test this premise.

On the other hand, recessive *BEST1* mutations from ARB patients provide a unique opportunity to analyze and connect the structure, function and physiological role of BEST1 in a 'clean' manner, as only the mutant BEST1 proteins are present in patients and all our experimental systems. By contrast, the co-existence of both WT and mutant BEST1 proteins in the cases of dominant mutations complicates the functional-structural analyses for several reasons: (1) as the pentameric bestrophin channels consist of five protomers, different numbers (0–5) of BEST1 mutant protomers could potentially be assembled to a BEST1 pentamer and impact the channel structure and function; (2) although the ratio of endogenous WT to mutant BEST1 proteins is key to determine the composition of pentameric BEST1 channels in patients, this critical factor cannot be determined by either western blot or immunostaining, as the BEST1-specific antibody cannot distinguish WT and mutant BEST1 proteins; (3) crystallographic studies with the BEST1 dominant mutant proteins only reflect homopentamers consisting of all five mutant protomers, but not heteropentamers with 1–4 mutant protomers; (4) it could be technically challenging to rescue phenotypes caused by dominant

mutations, and thus hard to draw a clear conclusion. Nevertheless, we are actively investigating *BEST1* dominant mutations using the pipelines established in this work with necessary modifications and cautions.

# Materials and methods

## Key resources table

| Reagent type (species) or resource | Designation | Source or reference | Identifiers | Additional information |
|---|---|---|---|---|
| gene (human) | BEST1 | PMID: 25324390 | | |
| gene (*Klebsiella pneumoniae*) | KpBest | PMID: 25324390 | | |
| strain, strain background (*E.coli*) | DH5alpha | other | | Laboratory of Wayne Hendrickson |
| strain, strain background (*E. coli*) | BL21 plysS | other | | Laboratory of Wayne Hendrickson |
| cell line (human) | HEK293 | other | RRID:CVCL_0045 | Laboratory of David Yule |
| transfected construct (human) | pEGFP-N1-BEST1 WT | PMID: 25324390 | | |
| transfected construct (human) | pEGFP-N1-BEST1 I201T | this paper | | Made from pEGFP-N1-BEST1 WT by site-directed mutagenesis |
| transfected construct (human) | pEGFP-N1-BEST1 P274R | this paper | | Made from pEGFP-N1-BEST1 WT by site-directed mutagenesis |
| biological sample (human) | skin cells | other | | New York Presbyterian Hospital |
| biological sample (human) | fetus eye samples | other | | New York Presbyterian Hospital |
| biological sample (human) | BEST1 WT iPSC-RPE | this paper | | Generated from donor skin cells by re-programming and differentiation |
| biological sample (human) | BEST1 I201T iPSC-RPE | this paper | | Generated from donor skin cells by re-programming and differentiation |
| biological sample (human) | BEST1 P274R iPSC-RPE | this paper | | Generated from donor skin cells by re-programming and differentiation |
| antibody | BESTROPHIN1 | Novus Biologicals NB300-164 | RRID:AB_10003019 | 1:200 |
| antibody | ZO-1 | Invitrogen 40–2200 | RRID:AB_2533456 | 1:500 |
| antibody | Alexa Fluor 488-conjugated IgG | Invitrogen A-11070 | RRID:AB_2534114 | 1:1000 |
| antibody | Alexa Fluor 555-conjugated IgG | Invitrogen A-21422 | RRID:AB_2535844 | 1:1000 |
| antibody | RPE65 | Novus Biologicals NB100-355 | RRID:AB_10002148 | 1:1000 |
| antibody | CRALBP | Abcam ab15051 | RRID:AB_2269474 | 1:500 |
| antibody | β-actin | Abcam ab8227 | RRID:AB_2305186 | 1:2000 |
| antibody | GFP | Invitrogen A6455 | RRID:AB_221570 | 1:5000 |
| antibody | SOX2, Tra-1–60, SSEA4, Nanog | Abcam ab109884 | | 1:200 |
| antibody | EEA1 | Fisher Scientific MA5-14794 | RRID:AB_10985824 | 1:200 |
| recombinant DNA reagent | pEG Bacmam | other | | Laboratory of Eric Gouaux |

*Continued on next page*

*Continued*

| Reagent type (species) or resource | Designation | Source or reference | Identifiers | Additional information |
|---|---|---|---|---|
| recombinant DNA reagent | pEG Bacmam-BEST1-GFP | this paper | | Made from pEG Bacmam by inserting BEST1-GFP |
| recombinant DNA reagent | BEST1-GFP Bacmam virus | this paper | | Produced from pEG Bacmam-BEST1-GFP by published protocols (*Goehring et al., 2014*) |
| recombinant DNA reagent | pMCSG7-10xHis-KpBest$^{\Delta C11}$ | PMID: 25324390 | | |
| recombinant DNA reagent | pMCSG7-10xHis-KpBest$^{\Delta C11}$ L177T | this paper | | Made from pMCSG7-10xHis-KpBest$^{\Delta C11}$ by site-directed mutagenesis |
| recombinant DNA reagent | pMCSG7-10xHis-KpBest$^{\Delta C11}$ P239R | this paper | | Made from pMCSG7-10xHis-KpBest$^{\Delta C11}$ by site-directed mutagenesis |
| sequence-based reagent | BEST1 I201T forward primer | this paper | | ACCCGGGACC CTATCCTGCT |
| sequence-based reagent | BEST1 I201T reverse primer | this paper | | GATAGGGTCCCGGG TTCGACCTCCAAGCCACG |
| sequence-based reagent | BEST1 P274R forward primer | this paper | | CGCGTCTTCAC GTTCCTGCAGTT |
| sequence-based reagent | BEST1 P274R reverse primer | this paper | | GAACGTGAAGAC GCGCACAACGAGGT |
| sequence-based reagent | KpBest L177T forward primer | this paper | | ACCAGCGACA TCACTTACGGGC |
| sequence-based reagent | KpBest L177T reverse primer | this paper | | AGTGATGTCGCT GGTCTTGCCCGCCTCCCG |
| sequence-based reagent | KpBest P239R forward primer | this paper | | CGGTTTGTCTCGGTC TTTATCTCTTACACC |
| sequence-based reagent | KpBest P239R reverse primer | this paper | | GACCGAGAC AAACCGCGTCA TGTA GTGCAGATCGC |
| peptide, recombinant protein | KpBest$^{\Delta C11}$ L177T | this paper | | Expressed from E. coli BL21 plysS, and purified by affinity and size-exclusion chromatography |
| peptide, recombinant protein | KpBest$^{\Delta C11}$ P239R | this paper | | Expressed from E. coli BL21 plysS, and purified by affinity and size-exclusion chromatography |
| commercial assay or kit | CytoTune-iPS 2.0 Sendai Reprogramming Kit | Thermo Fisher Scientific A16517 | | |
| commercial assay or kit | In-fusion Cloning Kit | Clontech 639645 | | |
| chemical compound, drug | mTeSR-1 medium | STEMCELL Technologies 5850 | | |
| chemical compound, drug | matrigel | CORNING 356230 | | |

*Continued on next page*

*Continued*

| Reagent type (species) or resource | Designation | Source or reference | Identifiers | Additional information |
|---|---|---|---|---|
| chemical compound, drug | nicotinamide | Sigma-Aldrich N0636 | | |
| chemical compound, drug | Activin-A | PeproTech 120–14 | | |
| software, algorithm | XDS | PMID: 20124692 | | |
| software, algorithm | Phaser | PMID: 19461840 | RRID:SCR_014219 | |
| software, algorithm | Phenix | PMID: 20124702 | RRID:SCR_014224 | |
| software, algorithm | Coot | PMID: 15572765 | RRID:SCR_014222 | |
| software, algorithm | PyMOL | http://www.pymol.org/ | RRID:SCR_000305 | |
| software, algorithm | Origin | http://www.originlab.com/index.aspx?go=PRODUCTS/Origin | RRID:SCR_014212 | |
| software, algorithm | MODELLER | PMID: 14696385 | RRID:SCR_008395 | |

## Generation of human iPSC

Primary fibroblasts cells from donors were reprogrammed into pluripotent stem cells using the Cyto-Tune-iPS 2.0 Sendai Reprogramming Kit (Thermo Fisher Scientific, A16517), and immunocytofluorescence assays were performed for scoring iPSC pluripotency following the previously published protocol (*Li et al., 2016*). In brief, a panel of antibodies (1:200, abcam, ab109884) against four standard pluripotency markers SOX2, Tra-1–60, SSEA4 and Nanog were applied to characterize the iPSCs from all the subjects enrolled in this study. Hoechst staining was applied to detect nuclei. Secondary antibodies were Alexa Fluor 488 conjugated goat anti-rabbit or Alexa Fluor 555 conjugated goat anti-mouse IgG (1:1,000; Life Technologies). Images for all antibody labels were taken under the same settings with fluorescence microscope (NIKON, Eclipse, Ts2R). All iPSC lines were maintained in mTeSR-1 medium (STEMCELL Technologies, 05850) and passaged every 3–6 days. The morphology and nuclear/cytoplasmic ratio of the iPSC lines were closely monitored to ensure the stability. To verify genome integrity, all the iPSC lines in this study were sent for karyotyping by G-banding at the Cell Line Genetics (Wisconsin, USA).

## Differentiation of iPSC into RPE

iPSC differentiation started at passage 4 for all iPSC lines. For differentiation, iPSC colonies were cultured to confluence in 6-well culture dishes (Costar, Corning, Corning, NY) pretreated with 1:50 diluted matrigel (CORNING, 356230) in differentiation medium consisting of Knock-Out (KO) DMEM (Thermo Fisher Scientific, 10829018), 15% KO serum replacement (Thermo Fisher Scientific, 10829028), 1% nonessential amino acids (Thermo Fisher Scientific, 11140050), 2 mM glutamine (Thermo Fisher Scientific, 35050061), 50 U/ml penicillin-streptomycin (Thermo Fisher Scientific, 10378016), and 10 mM nicotinamide (Sigma-Aldrich, N0636) for the first 14 days. During the 15th-28th days of differentiation, 100 ng/ml human Activin-A (PeproTech, 120–14) was supplemented into differentiation medium. From day 29, Activin-A supplementation was stopped until differentiation was completed. After 8–10 weeks, pigmented clusters were formatted and manually picked, then plated on matrigel-coated dishes in RPE culture medium as previous described (*Maminishkis et al., 2006*). They were cultured for another 6–8 weeks to allow them to form a functional monolayer for function assay. Besides well-established classical mature RPE markers RPE65, Bestrophin1 and CRALBP, two additional RPE markers, MITF and PAX6, were used for RPE fate validation. All the iPSC-RPE cells used in this study were at their passage 1. Mutations (P274R and I201T) in the mutant iPSC-RPEs were verified by sequencing.

## Cell lines

HEK293 cells were gifts from Dr. David Yule at University of Rochester. Although HEK293 is on the list of commonly misidentified cell lines maintained by the International Cell Line Authentication Committee, the HEK293 cells used in this study were authenticated by short tandem repeat (STR)

DNA profiling. No mycoplasma contamination was found. Low-passage-number HEK293 cells were maintained in DMEM supplemented with 10% FBS and 100 µg/ml penicillin-streptomycin.

## Immunofluorescence

Immunofluorescence staining was performed in all iPSC-RPE lines and human fetal RPE cells. Cells were washed with PBS and fixed in 4% paraformaldehyde for 45 min at room temperature. After washing with PBS twice, the cells were incubated in PBS with 0.1% Triton X-100% and 2% donkey serum for 45 min. Then, primary antibodies against BESTROPHIN-1 (1:200, Novus Biologicals, NB300-164), ZO-1 (1:500, Invitrogen Life Technologies, 40–2200) and EEA1 (1:200, Thermo Fisher Scientific, MA5-14794) were applied to each sample for 2 hr at room temperature. Alexa Fluor 488-conjugated and Alexa Fluor 555-conjugated IgG (1:1,000, Thermo Fisher Scientific) were used as secondary antibodies. Hoechst was used to detect the cell nuclei. Stained cells were observed by confocal microscopy (Nikon Ti Eclipse inverted microscope for scanning confocal microscopy, Japan).

## Electrophysiology

Whole-cell recordings of RPE and HEK cells were conducted 48–72 hr after splitting the cells or transfection, respectively, using an EPC10 patch clamp amplifier (HEKA Electronics) controlled by Patchmaster software (HEKA). Micropipettes were fashioned from 1.5 mm thin-walled glass with filament (WPI Instruments) and filled with internal solution containing (in mM): 130 CsCl, 1 $MgCl_2$, 10 EGTA, 2 MgATP (added fresh), 10 HEPES (pH 7.4), and $CaCl_2$ to obtain the desired free $Ca^{2+}$-concentration (maxchelator.stanford.edu/CaMgATPEGTA-TS.htm). Series resistance was typically 1.5–2.5 MΩ. There was no electronic series resistance compensation. External solution contained (in mM): 140 NaCl, 5 KCl, 2 $CaCl_2$, 1 $MgCl_2$, 15 glucose and 10 HEPES (pH 7.4). Whole-cell I-V curves were generated from a family of step potentials (−100 to +100 mV from a holding potential of 0 mV). Currents were sampled at 25 kHz and filtered at 5 or 10 kHz. Traces were acquired at a repetition interval of 4 s (*Yang et al., 2014a*).

Purified full length KpBest proteins were fused to planar lipid bilayers formed by painting a lipid mixture of phosphatidylethanolamine and phosphatidylcholine (Avanti Polar Lipids) in a 3:1 ratio in decane; across a 200 µm hole in polysulfonate cups (Warner Instruments) separating two chambers. The *trans* chamber (1.0 ml), representing the intra-SR (luminal) compartment, was connected to the head stage input of a bilayer voltage clamp amplifier. The *cis* chamber (1.0 ml), representing the cytoplasmic compartment, was held at virtual ground. Solutions were as follows (in mM): 150 NaCl, and 10 HEPES (pH 7.4) in the *cis* and *trans* solution. Purified proteins were added to the *cis* side and were fused with the lipid bilayer. Single-channel currents were recorded using a Bilayer Clamp BC-525D (Warner Instruments, LLC, CT), filtered at 1 kHz using a Low-Pass Bessel Filter 8 Pole (Warner Instruments, LLC, CT), and digitized at 4 kHz. All experiments were performed at room temperature (23 ± 2°C).

## Immunoblot analysis

Total cellular protein was extracted by M-PER mammalian protein extraction reagent buffer (Pierce, 78501) with proteinase inhibitor (Roche Diagnostics), and quantified by Bio-Rad protein reader. Protein samples (20 µg) were then separated on 10% Tris–Cl gradient gel and electro-blotted onto nitrocellulose membrane. The membranes were incubated in blocking buffer for 1 hr at room temperature, washed three times in PBS with 0.1% Tween for 5 min each, and incubated with primary antibody in blocking buffer overnight at 4°C. Primary antibodies against the following proteins were used for western blots: RPE65 (1:1,000 Novus Biologicals, NB100-355), BESTROPHIN-1 (1:500 Novus Biologicals, NB300-164), CRALBP (1:500 Abcam, ab15051), β-actin (1:2,000 Abcam, ab8227), and GFP (1:5,000 Invitrogen, A6455). Mouse and rabbit secondary antibodies were obtained from Santa Cruz and used at a concentration of 1: 5000.

## Virus

WT BEST1-GFP expressed from a BacMam baculoviral vector was made as previously described (*Goehring et al., 2014*), and was added into RPE culture 24 hr after splitting the cells (MOI = 100).

## cDNA cloning

P237R and L177T KpBest$^{\Delta C11}$ have 11 residues truncated from the C-terminus of wild-type KpBest. The wild-type BEST1 (synthesized by Genscript), was amplified using polymerase chain reaction (PCR), and was subcloned into a pEGFP-N1 mammalian expression vector. C-terminus truncated KpBest and point mutations of KpBest and BEST1 were made using the In-fusion Cloning Kit (Clontech). All clones were verified by sequencing.

## Transfection

For electrophysiology experiments, HEK293 cells cultured in 6 cm tissue culture dishes were transiently transfected with the indicated BEST1 (6 µg) and T antigen (2 µg), using the calcium phosphate precipitation method. Cells were washed with PBS 4–8 hr after transfection and maintained in supplemented DMEM, and replated onto fibronectin-coated glass coverslips 24 hr after transfection (*Yang et al., 2013*).

## Protein production and purification

BL21 plysS cells were gifts from Dr. Wayne Hendrickson. For scaling up, transformed BL21 plysS cells were grown at 37°C in TB media to OD 0.6–0.8 after being inoculated with 1% of the overnight culture. The culture was induced with 0.4 mM IPTG and continued to grow at 37°C for another 4 hr.

BL21 plysS cells expressing targeted proteins were harvested by centrifugation and stored at −80°C before use. Cells were resuspended in a buffer containing 50 mM HEPES (pH 7.8) and 200 mM NaCl and lysed using a French Press with two passes at 15–20,000 psi. Cell debris was removed by centrifugation at 10,000 g for 20 min, and the membrane fraction was isolated from that supernatant by ultra-centrifugation at 150,000 g for 1 hr.

The membrane fraction was homogenized in a solubilization buffer containing 50 mM HEPES (pH 7.8) and 300 mM NaCl, and incubated with a final concentration of 0.05% (w/v) DDM for 1 hr at 4°C. The non-dissolved matter was removed by ultracentrifugation at 150,000 g for 30 min, and the supernatant was loaded to a 5 ml Hitrap Ni$^{2+}$-NTA affinity column (GE Healthcare), pre-equilibrated with the same solubilization buffer supplemented with 0.05% DDM. After 20 column volume buffer wash, the protein was eluted with 500 mM imidazole in the solubilization buffer. The 10-His tags were removed by adding super TEV at 1:1 mass ratio and incubating at 4°C for 30 min. Tag removal was confirmed by SDS-PAGE, and the resulting sample was concentrated to approximately 10 mg/ml. Preparative size-exclusion chromatography was carried out on a Superdex-200 column for further purification, including removal of TEV protease and the cleaved tag. The gel-filtration buffer contained 40 mM HEPES (pH 7.8), 200 mM NaCl, 0.1 mM Tris [2-carboxyethyl] phosphine (TCEP), and 2 × CMC of detergent DDM.

## Crystallization and data collection

Purified protein was concentrated to ~10 mg/ml. Crystals were all grown at 20°C using the sitting-drop vapor diffusion method. The condition contained 0.05 M zinc acetate, 6% v/v ethylene glycol, 0.1 M sodium cacodylate, pH 6.0, and 6.6 % w/v PEG 8000. Cryoprotection was achieved by adding 20% ethylene glycol to the crystallization solution. High resolution native data set from a single L177T KpBest$^{\Delta C11}$ crystal was collected at APS (Argonne National Laboratory) beamline 24-ID-E.

## Statistics

### Electrophysiological data and statistical analyses

Whole-cell clamp data were analyzed off-line using Patchmaster (HEKA), Microsoft Excel and Origin software. Statistical analyses were performed in Origin using built-in functions. Statistically significant differences between means ($p<0.05$) were determined using Student's *t* test for comparisons between two groups, and one-way ANOVA and Bonferroni *post hoc* analyses between more than two groups. Data are presented as means ± s.e.m (*Yang et al., 2007*).

### Structure determination and refinement

The x-ray data set on L177T KpBest$^{\Delta C11}$ was processed using XDS (*Kabsch, 2010*) via the RAPD system of APS NE-CAT. The structure was solved using WT KpBest$^{\Delta C11}$ structure (PDB code: 4WD8) as a search model during molecular replacement, carried out using the program Phaser (*McCoy et al.,*

*2007*) as implemented in the program Phenix suite (*Adams et al., 2010*). Model building and refinement were carried out using the programs Coot (*Emsley and Cowtan, 2004*) and Phenix suite (*Adams et al., 2010*). The statistics for the diffraction data and refinement are summarized in *Figure 8—source data 1*.

## Homology modeling of human BEST1

Homology models for BEST1 were generated using MODELLER (*Fiser and Sali, 2003*). All figures were made in PyMOL.

## Data and software availability

The data reported in this paper are tabulated in *Figure 8—source data 1*, and deposited to the Protein Data Bank with access codes listed in *Figure 8—source data 1*.

## Study approval

### Patients and clinical analysis

Patient 1 is a 12-year-old otherwise healthy boy, and patient 2 is a 72-year-old otherwise healthy man. Two BEST1-mutant patients underwent a complete ophthalmic examination by a retinal physician in the Department of Ophthalmology, Columbia University Medical Center/New York Presbyterian Hospital. This included best-corrected visual acuity, slit-lamp biomicroscopy, and dilated funduscopy. Both of the patients underwent color fundus photography, optical coherence tomography (OCT) and electroretinogram (ERG) (*Kohl et al., 2015*; *McCulloch et al., 2015*). Skin biopsy samples were obtained from patients and healthy control donors, and processed and cultured as previously described (*Li et al., 2016*). Patients and the parent/legal guardian of patient 1 provided written informed consent for all procedures, which were approved by Columbia University Institutional Review Board (IRB) protocol AAAF1849.

### Fetal human RPE isolation and culture

Human RPE cells were isolated and cultured from human fetal eye samples (13 to 14 weeks old) obtained from Department of OB/GYN, New York Presbyterian Hospital (Protocol number: IRB-AAAO1804 and IRB-AAAQ7782), as described previously (*Sonoda et al., 2009*). In brief, the eyeball with anterior portions and vitreous was removed and then incubated in 2% dispase at 37°C for 45 min. Next, the RPE layer was separated from the choroid layer and transferred to a 15 ml conical tube containing 0.25% trypsin-EDTA. Then the tube was incubated in 37°C water bath for 10 min. After centrifugation at 0.8 rpm for 4 min, the cell pellet was resuspended in RPE medium and plated on a matrigel coated petri dish. All the fetal RPE cells used in this study were at their passage 1.

## Acknowledgements

We thank Henry Colecraft for comments on the paper, Anne R Davis for providing human globe samples, David Yule for HEK293 cells, David Yule and Yu (Julie) Zhang for help in taking confocal images, Qun Liu and Min Su for help in generating the human model, Yota Fukuda for suggestions on model analysis and staff at the Advanced Photon Source (APS) beamline 24-ID-E for their assistance in data collection. This work used NE-CAT beamlines (GM103403) at the APS (DE-AC02-06CH11357). SC was supported by Sun Yat-Sen University "100 Top Talents Program (II) and the National Natural Science Foundation of China (Grant No. 31770801). SHT was supported by Barbara and Donald Jonas Family Fund, and Research to Prevent Blindness. This work was supported by University of Rochester start-up funding to TY.

## Additional information

### Funding

| Funder | Grant reference number | Author |
| --- | --- | --- |
| National Natural Science Foundation of China | 31770801 | Shoudeng Chen |

| Sun Yat-sen University | 100 Top Talents Program (II) | Shoudeng Chen |
| Barbara and Donald Jonas Family Fund | | Stephen H Tsang |
| Research to Prevent Blindness | | Stephen H Tsang |
| University of Rochester | Start-up funding | Tingting Yang |

The funders had no role in study design, data collection and interpretation, or the decision to submit the work for publication.

### Author contributions
Yao Li, generated iPSC-RPE and fhRPE cells, performed confocal microscopy and western blot, and analyzed data; Yu Zhang, maintained RPE culture, performed western blot, bacterial protein expression, purification and crystallization, made the virus and wrote the paper; Yu Xu, helped generate iPSC-RPE and fhRPE cells, and performed western blot; Alec Kittredge, generated constructs in Figures 5, 7 and 8; Nancy Ward, helped with RPE culture and made the virus; Shoudeng Chen, analyzed diffraction data; Stephen H Tsang, cared for BEST1 patients and performed skin biopsy; Tingting Yang, designed experiments, performed patch clamp and lipid bilayer experiments, analyzed data, made figures and wrote the paper

### Author ORCIDs
Tingting Yang (iD) http://orcid.org/0000-0002-5220-588X

### Ethics
Human subjects: Two BEST1-mutant patients (12-year-old and 72-year-old, respectively) underwent a complete ophthalmic examination by a retinal physician in the Department of Ophthalmology, Columbia University Medical Center/New York Presbyterian Hospital. This included best-corrected visual acuity, slit-lamp biomicroscopy, and dilated funduscopy. Both of the patients underwent color fundus photography, optical coherence tomography (OCT) and electroretinogram (ERG). Skin biopsy samples were obtained from patients and healthy control donors, and processed and cultured as previously described (Li et al, 2016). Patients and the parent/legal guardian of patient 1 (12-year-old) provided written informed consent for all procedures, which were approved by Columbia University Institutional Review Board (IRB) protocol AAAF1849. Human RPE cells were isolated and cultured from human fetal eye samples (13 to 14 weeks old) obtained from Department of OB/GYN, New York Presbyterian Hospital (Protocol number: IRB-AAAO1804 and IRB-AAAQ7782), as described previously (Sonoda et al, 2009).

### Decision letter and Author response
Decision letter https://doi.org/10.7554/eLife.29914.027
Author response https://doi.org/10.7554/eLife.29914.028

## Additional files
### Supplementary files
• Transparent reporting form
DOI: https://doi.org/10.7554/eLife.29914.022

### Major datasets
The following dataset was generated:

| Author(s) | Year | Dataset title | Dataset URL | Database, license, and accessibility information |
|---|---|---|---|---|
| Zhang Y, Chen S, Yang T | 2017 | Crystal structure of a bacterial Bestropin homolog from Klebsiella pneumoniae with a mutation L177T | http://www.rcsb.org/pdb/explore/explore.do?structureId=5X87 | Publicly available at the RCSB PDB website (accession no. 5X87) |

The following previously published dataset was used:

| Author(s) | Year | Dataset title | Dataset URL | Database, license, and accessibility information |
|---|---|---|---|---|
| Yang T, Liu Q, Hendrickson WA | 2014 | Crystal structure of a bacterial Bestrophin homolog from Klebsiella pneumoniae | http://www.rcsb.org/pdb/explore/explore.do?structureId=4WD8 | Publicly available at the RCSB PDB website (accession no: 4WD8) |

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
