## [Decision Letter]

Thank you for submitting your article "Patient-specific mutations impair an essential role of BESTROPHIN1" for consideration by *eLife*. Three experts reviewed your manuscript, and their assessments, together with my own (Reviewing Editor), form the basis of this letter. The evaluation has been overseen by Richard Aldrich as the Senior Editor.

As you will see, all of the reviewers were impressed with the importance and novelty of your work, but they also had a wide variety of critiques. I am including the three reviews at the end of this letter, as there are many specific and useful suggestions in them.

In further discussions among the reviewers, the major questions that arose focused on:

1) Potential variability among different cell lines from the same donor – i.e. how reproducible are the data obtained from different cell lines?

2) The distinction between *Best1* current and other non-*Best1* currents (e.g. VRAC) – i.e. a fuller characterization of the rescued current.

We appreciate that the reviewers' comments cover a very broad range of suggestions for improving the manuscript. Please use your best judgment in deciding which of these can be accommodated in a reasonable period of time. We look forward to receiving your revised manuscript.

*Reviewer #1:*

This study provides the first direct evidence that the endogenous human Bestrophin 1 is responsible for calcium-dependent chloride current in human RPE cells and that disease-associated mutations in human Bestrophin 1 abolishes this current in human RPE cells. This study also offers the most direct evidence that Best disease is a channelopathy caused by a defective calcium-activated chloride channel and overcomes the difficulty of previous mouse models. This achievement warrants its publication in *eLife*. This reviewer does not suggest any further experiments, but suggests improvements in the Discussion and presentation. Here are a few high relevant points that can be discussed to improve the manuscript:

1) What are the differences in locations between the recessive mutations and dominant mutations in Bestrophin 1? Can the structural model predict or make sense of this functional difference? Since the data showed that the recessive mutants are still expressed at the protein level, it is surprisingly that the mutant subunit does not interfere with the function of the wild-type subunits in the pentamer (it does, it would be a dominant mutant).

2) The authors write "in any case, our results do not exclude the role of TMEM16A/TMEM16B in regulating Ca^2+^-dependent Cl^-^ current in human RPE." However, the authors presented data showing that RPE cells with the mutant Bestrophin 1 lacks any Ca^2+^-dependent Cl^-^ current. This data does exclude the role of TMEM16A/TMEM16B in regulating Ca^2+^-dependent Cl^-^ current in human RPE.

3) The authors state "most of the *BEST1* patient-specific mutations are dominant, so that the mutant *BEST1* alleles in these cases may be gain-of-function as oppose to the loss-of-function alleles in ARB patients." There is only evidence to suggest that the dominant mutations are loss-of-function mutations. Is there any evidence that the dominant mutations cause a gain of function? This is also related to discussion point 1 above.

4) This study's demonstration that Bestrophin 1 is responsible for the calcium-dependent chloride current in human RPE cells makes sense physiologically and clinically given the fact that EOG is used routinely as a clinical tool to detect Best disease in human patients. Can the authors explain the difficulty to demonstrate this activity in mouse models and the lessons that can be learned on drawing conclusions using mouse models, which are still the most common used models?

*Reviewer #2:*

1) N=1 Patient iPSC-RPE I201T (mild phenotype), N=1 Patient iPSC-RPE P274R (extreme phenotype). These findings should be replicated using more clones per iPSC line. Low N number cannot mitigate donor/clonal variability, which has been seen to be quite high in patient and control iPSC samples.

2) The authors state: "Here for the first time, we directly measured Ca^2+^- dependent Cl currents on the plasma membrane of human RPEs by whole-cell patch clamp" This statement raises a critical question: Where is *Best1* localized "on the plasma membrane" in these human RPEs?

• Is it the apical or basolateral membrane? The authors stain for ZO-1 and *Best1*, but ZO-1 is an apical membrane marker. Immunostaining is low resolution, but it shows that *BEST1* does *not* co-localize with ZO1. It seems cytoplasmic. Authors can rule out cytoplasmic localization using ER and Endosome markers. In other systems, basolateral localization is indicated (Johnson et al., 2017). Typical apical membrane markers for human RPE are ezrin or N/K ATPase and collagen IV is a marker for the basolateral membrane.

3) The reduced light peak in patients is discussed extensively but the question remains – is the calcium activated chloride current measured in the present study the in vivo LP response of the RPE? The authors state: "Our results provide definitive evidence that the CaCC activity of *Best1*"

• It would be important to know if the Cl channel currents (e.g. Figure 1) attributed to *Best1* can be pharmacologically blocked by appropriately specific inhibitors at the basolateral (or apical) membranes. The blocker specificity of NFA (Figure 1—figure supplement 1; Figure 5—figure supplement 1) is assumed but it is variable from one system to the next. For example, in native bovine RPE NFA increases basolateral membrane Cl conductance and the resultant Cl current is blocked by basal DIDS as measured by intracellular recording and by decreased 36Cl flux. In bovine RPE intracellular Cl was significantly decreased by NFA. By comparison, the current authors (Moshfegh et al., 2016) showed that elevated Ca (physiological?) also decreased intracellular Cl activity in iPSC-derived RPE as inferred from the initial decrease in measured YFP fluorescence. In the latter experiments the intracellular Ca increase could be acting at the plasma membrane or at locations inside the cell and the change in Cl activity could for example be mediated by apical membrane Na/K/2Cl cotransporters, or basolateral membrane HCO_3_/Cl exchangers (but see 4 below), or other Ca-mediated transporters/channels. It would be worthwhile measuring the intracellular Cl activities (YFP) in the presence and absence of NFA. It would be quite useful to provide a calibration for the Cl activity baseline and subsequent Ca-induced changes for the cells used in the present study (e.g. Watts et al. PLoS ONE 2012).

4) Also, in the present experiments, the use of HEPES buffer (subsection “Electrophysiology”, first paragraph) is an impediment to understanding how these responses might be connected to in vivo or in vitro LP responses (https://www.ncbi.nlm.nih.gov/pubmed/7153919). The absence of HCO_3_ buffer significantly deactivates a variety of apical and basolateral membrane transporters that control cell pH, Na, Cl, K, and volume in and around the RPE. For example, the apical membrane NaHCO_3_ co-transporter is electrogenic and if absent from these cells would reduce apical and basolateral membrane potential potentials, ion gradient's, and volume regulation and alter net fluid transport. These outcomes would obscure the possible connection between Ca-dependent currents measured in vitro and the in vivo events following light onset and the subsequent generation of the LP.

5) Authors do not provide mechanistic evidence regarding the driving forces that generate the calcium activated chloride channel. Is it calcium from internal stores (ER, melanosomes, mitochondria?) or is it external calcium? What happens if the apical membrane L-type calcium channels are blocked? What happens if ER calcium is dumped and reuptake inhibited prior to recording?

6) As noted by the authors most of the *BEST1* mutations are dominant. The measurement of aberrant Ca-dependent Cl current in RPE from that prevalent cohort of patents should also have been a priority. The authors showed that the two ARB patients they studied have significantly altered ERG a- and b-waves, indicating that the retinal input to generate the RPE LP is very likely abnormal. In contrast, autosomal dominant Best Vitelliform Macular Dystrophy (BVMD) patients exhibit normal ERGs and thus subsequent abnormal RPE LP can be more clearly analyzed in terms of RPE-specific abnormalities.

7) One area of focus going forward could involve the ATP-induced activation of RPE apical membrane purinergic receptors that significantly increase cell calcium levels, and basolateral membrane Cl transport mediated fluid transport across the RPE. Activation of apical membrane alpha1 adrenergic receptors produces a similar sequence of events. A comparison of Ca-activated Cl currents in iPS-derived RPE from patients with recessive and dominant mutations relative to WT may provide a basis for functionally distinguishing these two genetically different types of patient.

Possible title modification: Characterization of Bestrophin1- mediated calcium activated chloride currents in human RPE.

9) Karyotyping, iPSC characterization, and iPSC-RPE validation data information (beyond RPE65 and CRALBP in iPSC-derived RPE) – should be included in text. Please specify the passage number for primary cultures of human fetal RPE.

10) In the manuscript, please provide sequencing information that verifies the original mutation after reprogramming into iPSC and differentiation into RPE.

11) IPSC were differentiated at passage 4. What criteria were used to ensure the stability of lines prior to differentiation?

*Reviewer #3:*

This is an impressive study that addresses a long-standing question in the ion channel and visual science fields. Mutations in the Ca-activated Cl channel *BEST1* cause a spectrum of retinopathies known as bestrophinopathies, but the mechanisms have remained obscure at least partly because mouse models do not faithfully reproduce human. A hallmark of bestrophinopathies is the absence of the light peak in the electrooculogram. While it has long been suspected that *BEST1* is the channel responsible for the light peak, evidence has been lacking. This paper addresses the question by (1) identifying 2 patients with *BEST1* mutations and characterizing their phenoptype, (2) showing that RPE cells differentiated from iPSCs derived from these bestrophinopathy patients have defective Ca-activated Cl currents, (3) showing that HEK cells expressing the patient mutations have defective Cl currents, and (4) solving the crystal structure of a *BEST1* homolog (KpBest) with one of the patient mutations and recording single channel currents. This very powerful combination of approaches provides a strong argument that *BEST1* is the Ca-activated Cl channel responsible for the light peak.

Despite the strength of the multi-dimensional approach, each aspect of the study has serious weaknesses.

1) The evidence that the ionic currents in iPSC-RPE cells that are interpreted as encoded by *BEST1* is weak. First, the current traces have the characteristic unmistakable appearance of volume-regulated anion channels (VRAC) and they do not resemble hBEST1 currents expressed in HEK cells (as can be seen comparing Figure 1 and Figure 5). This could mean that the currents are mediated VRAC (LRRC8) which are somehow regulated by *BEST1*. Second, the experiments the authors perform to conclude these are *BEST1* currents are superficial. There is no data in this paper showing their anion selectivity. The use of NFA as a Ca-activated Cl channel blocker is bogus: NFA is more potent at blocking VRAC and TMEM16A than *BEST1* (PMID:28620305;PMID:25078708). The authors must rule out the possibility that the currents they are studying are VRAC currents. This could involve knockdown of LRRC8A. Another approach to distinguish between VRAC and *BEST1*, the authors might measure taurine permeability (taurine is zwitterionic so taurine currents can be measured at the appropriate pH). This is a crucial point. The authors might argue that VRAC is not Ca-activated, but VRAC is clearly Ca-regulated.

2) The immunofluorescence images showing *BEST1* on the cell surface are not convincing. In Figure 1 and Figure 2, ZO-1 is clearly on the cell surface, but most of *BEST1* is intracellular. It is not clear that any *BEST1* co-localizes with ZO-1. The authors must use deconvolution or super-resolution imaging and perform statistical analysis to show co-localization. This should be supplemented with cell surface biotinylation. Also, because the CaCC that generates the light peak is presumed to be on the basolateral membrane, the authors should determine whether *BEST1* is trafficked properly in these cultures. In Figure 4 the localization of *BEST1* is not obviously different from control. The level of expression seems spotty in the P274R mutant, but some cells look like they express at control levels. ZO-1 is very clearly disrupted in the I201T mutant, but the authors do not comment on this. It is also curious that overexpressed *BEST1* localizes more strongly to the plasma membrane than does the endogenous protein detected by antibody. This raises questions about the validity of the antibody. Has it been knockout-verified in IF?

3) While the homology model is informative, the crystal structure of the KpBest L177T mutant does not provide any important insights. The KpBest L177T mutant is meant to model the hBEST1 I201T mutant. Not only is the mutated residue non-identical but of the 10 amino acids on either side of L177 (residues 167-187) only 3 amino acids are identical and not many more are structurally similar. Furthermore, KpBest is not a Cl channel, but a cation channel. It is puzzling why the authors did not use chicken *BEST1,* which has also been successfully crystalized and is very closely related to hBEST1. The shift in the helices shown in Figure 8 does not obviously provide any mechanistic insights. A more detailed analysis of the effects of this mutation on the channel pore is required, but since this is a cation channel, I am not sure that the conclusions would be relevant. The single channel analysis supports the authors' conclusions but would be more valuable if they were performed with cBest1. How many times was this experiment repeated?

4) No methods are provided about how the ERGs were measured. How long were patients dark-adapted? What was the light stimulus intensity/duration, etc.? More explanation is required about the "age-matched controls". Were the controls measured co-temporaneously and how were they selected? Although the literature is a little ambiguous about ERG changes with age, I am a little surprised that normal teenagers have b-waves twice as large as a normal 72-year old. Not only should the authors provide more information about their own controls, but should compare their results to the published literature. At least one study shows no difference in b-wave amplitude between 20-39 year-olds and 60-82 year-olds (Doc Ophthalmol (2011) 122: 177.)

5) Of the two mutations studied, the P274R is clearly the most dramatic. However, the impact of the cellular data are weakened because there are no electro-oculogram data presented for this patient. The authors should show current traces for the rescued cells and should show that the rescued currents have the same Ca sensitivity as WT. While the authors state that "all patients display reduced LP", it is not clear exactly what this means. There are cases in the literature describing patients with *BEST1* mutations that have normal LPs.

6) In most experiments, only 4-5 cells were tested. Were these cells all from the same RPE colony derived from the same iPSC culture, or are they from different RPE colonies? If these data are from a single colony, how can one be certain that the difference in ionic currents is not explained by technical differences between colonies?

---

## [Author Response]

1) Potential variability among different cell lines from the same donor – i.e. how reproducible are the data obtained from different cell lines?

We appreciate the concern about potential variability among different cell lines from the same donor. We have now included additional clones for the WT and I201T mutant in the new Figure 1—source data 1. In both cases, the Ca^2+^-dependent Cl^-^ current amplitudes in two distinct clonal iPSC-RPEs from the same donor are next to identical, indicating the reproducibility and reliability of our data. We currently do not have mature P274R iPSC-RPE cells from a second clone. As the maturation time of iPSC-RPE varies from clone to clone, we are uncertain when an additional P274R iPSC-RPE line(s) may become available for testing within the next few months. As an alternative, we have now included data from P274R iPSC-RPE generated by a different set of differentiation in Figure 1—source data 1, which displayed consistent results.

2) The distinction between Best1 current and other non-Best1 currents (e.g. VRAC) – i.e. a fuller characterization of the rescued current.

We have now included the rescued current amplitude in P274R iPSC-RPE cells across a range of [Ca^2+^]_i_ in Figure 4 rescued trace exemplar in the new Figure 4—figure supplement 1. The pattern of Ca^2+^ response was similar in WT and rescued P274R iPSC-RPEs (EC_50_ 455 nM vs. 446 nM, Figure 4), the rescued current trace had the appearance of WT RPE endogenous current, and the rescued current was inhibited by NFA to a similar level as the WT RPE endogenous current, indicating that the recorded currents in WT and rescued P274R iPSC-RPEs are both Ca^2+^-dependent Cl^-^ currents. This Ca^2+^-dependent Cl^-^ current is apparently dependent on BEST1, as the null phenotype in P274R iPSC-RPE was fully rescued by complementation with WT BEST1. As BEST1 is a Ca^2+^-activated Cl^-^ channel specifically expressed in the RPE surface, the simplest model according to the principle of Occam’s Razor is that BEST1 conducts this Ca^2+^-dependent Cl^-^ current. This model is further supported by the direct correlation between the structural/functional deficiency of BEST1 channels and the severity of the clinical and cellular phenotypes. However, we cannot rule out the possibility that BEST1 somehow plays an indispensable role in regulating or cooperating with other anion channels such as TMEM16A, TMEM16B or other Ca^2+^-dependent VRAC. In fact, BEST1 itself has been suggested to function as a VRAC (Fischmeister and Hartzell, 2005; Qu and Hartzell, 2008). To conclusively exclude or include additional player(s) in mediating Ca^2+^-dependent Cl^-^ current in RPE, one needs to perform individual knockout of each candidate gene, which is beyond the scope of this work. The focus of this work is on Ca^2+^-dependent Cl^-^ current in RPE. VRAC current in RPE and the putative role of BEST1 as a VRAC are important but separate topics, which need to be carefully addressed in future studies.

We appreciate that the reviewers' comments cover a very broad range of suggestions for improving the manuscript. Please use your best judgment in deciding which of these can be accommodated in a reasonable period of time. We look forward to receiving your revised manuscript.Reviewer #1:1) What are the differences in locations between the recessive mutations and dominant mutations in Bestrophin 1? Can the structural model predict or make sense of this functional difference? Since the data showed that the recessive mutants are still expressed at the protein level, it is surprisingly that the mutant subunit does not interfere with the function of the wild-type subunits in the pentamer (it does, it would be a dominant mutant).

Mutations causing ARB are mostly located outside of the exons that usually harbor vitelliform macular dystrophy–associated dominant mutations (Fung et al., 2015). Unfortunately, it is still hard to predict the functional differences between dominant and recessive mutations based on their locations on the structural model. For the P274R mutation, our immunostaining results showed that the mutant protein is not localized on the plasma membrane, suggesting that the mutant subunit cannot traffic onto the plasma membrane to interfere with the WT subunit. For the I201T mutation, as the structural alteration is subtle even in the mutant homopentamer (Figure 8), it may require certain numbers of I201T mutant protomers in the heteropentamers to cause a significant phenotype in the channel function. Moreover, whether or not I201T mutant protomers can be properly assembled to form heteropentamers with WT protomers is unknown.

2) The authors write "in any case, our results do not exclude the role of TMEM16A/TMEM16B in regulating Ca^2+^-dependent Cl^-^ current in human RPE." However, the authors presented data showing that RPE cells with the mutant Bestrophin 1 lacks any Ca^2+^-dependent Cl^-^ current. This data does exclude the role of TMEM16A/TMEM16B in regulating Ca^2+^-dependent Cl^-^ current in human RPE.

We agree with the reviewer that our results indicate an indispensable role of BEST1 in mediating Ca^2+^-dependent Cl^-^ current in human RPE. As we stated in response to major question #2, the idea that BEST1 conducts this current, independent of any other channels, fits well with the principle of Occam’s Razor. However, we cannot officially rule out the possibility that BEST1 somehow cooperates with or regulates other Ca^2+^-regulated Cl^-^ channels (e.g. TMEM16A, TMEM16B or Ca^2+^-dependent VRAC). We have revised the manuscript: “our results do not completely exclude the role of other CaCCs in contributing to Ca^2+^-dependent Cl^-^ current in human RPE”.

3) The authors state "most of the BEST1 patient-specific mutations are dominant, so that the mutant BEST1 alleles in these cases may be gain-of-function as oppose to the loss-of-function alleles in ARB patients." There is only evidence to suggest that the dominant mutations are loss-of-function mutations. Is there any evidence that the dominant mutations cause a gain of function? This is also related to discussion point 1 above.

We thank the reviewer for pointing this out. We have revised this sentence to “… so that the mutant *BEST1* alleles in these cases may be more functionally defective and/or structurally disruptive compared to recessive mutant alleles in ARB patients.”

4) This study's demonstration that Bestrophin 1 is responsible for the calcium-dependent chloride current in human RPE cells makes sense physiologically and clinically given the fact that EOG is used routinely as a clinical tool to detect Best disease in human patients. Can the authors explain the difficulty to demonstrate this activity in mouse models and the lessons that can be learned on drawing conclusions using mouse models, which are still the most common used models?

There are significant species-specific differences between mouse and human RPE cells. For instance, only 3% of human RPE cells are binucleate, in contrast to 35% in mice (Volland et al., 2015). In regard to mutation-caused retinal diseases, the differences between these two species are even more obvious. Besides *BEST1*, defects in the *TIMP3* and *EFEMP1* genes both cause much more severe eye phenotypes in humans as compared to knockout/knock-in mice. We agree with the reviewer that conclusions should be drawn with cautions on the differences between species and the advantages/limitations of different experimental systems.

Reviewer #2:1) N=1 Patient iPSC-RPE I201T (mild phenotype), N=1 Patient iPSC-RPE P274R (extreme phenotype). These findings should be replicated using more clones per iPSC line. Low N number cannot mitigate donor/clonal variability, which has been seen to be quite high in patient and control iPSC samples.

Please refer to response to major question #1.

2) The authors state: "Here for the first time, we directly measured Ca^2+^- dependent Cl currents on the plasma membrane of human RPEs by whole-cell patch clamp" This statement raises a critical question: Where is Best1 localized "on the plasma membrane" in these human RPEs?• Is it the apical or basolateral membrane? The authors stain for ZO-1 and Best1, but ZO1 is an apical membrane marker. Immunostaining is low resolution, but it shows that BEST1 does not co-localize with ZO1. It seems cytoplasmic. Authors can rule out cytoplasmic localization using ER and Endosome markers. In other systems, basolateral localization is indicated (Johnson et al., 2017). Typical apical membrane markers for human RPE are ezrin or N/K ATPase and collagen IV is a marker for the basolateral membrane.

As the reviewer suggested, we co-stained BEST1 with the endosome marker EEA1. The results show that BEST1 does not co-localize with EEA1 (Figure 1—figure supplement 1), ruling out the cytoplasmic localization of BEST1. We agree that it is critical to determine whether BEST1 localizes on the apical or basal membrane in human RPE. However, in our system, RPE cells are directly attached to the culturing dish, so that the polarity of the cells may not best represent the status of RPE cells in vivo where supporter cells and a more sophisticated 3-D environment are present. Consistent with this notion, we have tried to use collagen IV as a basolateral membrane marker to stain our RPE culture, but the result was unclear. Overall, we can only conclude that BEST1 is localized on the plasma membrane of human RPE cells in our system, but cannot further distinguish between apical and basolateral localizations. We are excited to investigate this question in follow-up studies under more physiological conditions such as in vitro 3-D RPE cultures.

3) The reduced light peak in patients is discussed extensively but the question remains – is the calcium activated chloride current measured in the present study the in vivo LP response of the RPE? The authors state: "Our results provide definitive evidence that the CaCC activity of Best1"• It would be important to know if the Cl channel currents (e.g. Figure 1) attributed to Best1 can be pharmacologically blocked by appropriately specific inhibitors at the basolateral (or apical) membranes. The blocker specificity of NFA (Figure 1—figure supplement 1; Figure 5—figure supplement 1) is assumed but it is variable from one system to the next. For example, in native bovine RPE NFA increases basolateral membrane Cl conductance and the resultant Cl current is blocked by basal DIDS as measured by intracellular recording and by decreased 36Cl flux. In bovine RPE intracellular Cl was significantly decreased by NFA. By comparison, the current authors (Moshfegh et al., 2016) showed that elevated Ca (physiological?) also decreased intracellular Cl activity in iPSC-derived RPE as inferred from the initial decrease in measured YFP fluorescence. In the latter experiments the intracellular Ca increase could be acting at the plasma membrane or at locations inside the cell and the change in Cl activity could for example be mediated by apical membrane Na/K/2Cl cotransporters, or basolateral membrane HCO_3_/Cl exchangers (but see 4 below), or other Ca-mediated transporters/channels. It would be worthwhile measuring the intracellular Cl activities (YFP) in the presence and absence of NFA. It would be quite useful to provide a calibration for the Cl activity baseline and subsequent Ca-induced changes for the cells used in the present study (e.g. Watts et al. PLoS ONE 2012).4) Also, in the present experiments, the use of HEPES buffer (subsection “Electrophysiology”, first paragraph) is an impediment to understanding how these responses might be connected to in vivo or in vitro LP responses (https://www.ncbi.nlm.nih.gov/pubmed/7153919). The absence of HCO_3_ buffer significantly deactivates a variety of apical and basolateral membrane transporters that control cell pH, Na, Cl, K, and volume in and around the RPE. For example, the apical membrane NaHCO_3_ co-transporter is electrogenic and if absent from these cells would reduce apical and basolateral membrane potential potentials, ion gradient's, and volume regulation and alter net fluid transport. These outcomes would obscure the possible connection between Ca-dependent currents measured in vitro and the in vivo events following light onset and the subsequent generation of the LP.5) Authors do not provide mechanistic evidence regarding the driving forces that generate the calcium activated chloride channel. Is it calcium from internal stores (ER, melanosomes, mitochondria?) or is it external calcium? What happens if the apical membrane L-type calcium channels are blocked? What happens if ER calcium is dumped and reuptake inhibited prior to recording?6) As noted by the authors most of the BEST1 mutations are dominant. The measurement of aberrant Ca-dependent Cl current in RPE from that prevalent cohort of patents should also have been a priority. The authors showed that the two ARB patients they studied have significantly altered ERG a- and b-waves, indicating that the retinal input to generate the RPE LP is very likely abnormal. In contrast, autosomal dominant Best Vitelliform Macular Dystrophy (BVMD) patients exhibit normal ERGs and thus subsequent abnormal RPE LP can be more clearly analyzed in terms of RPE-specific abnormalities.7) One area of focus going forward could involve the ATP-induced activation of RPE apical membrane purinergic receptors that significantly increase cell calcium levels, and basolateral membrane Cl transport mediated fluid transport across the RPE. Activation of apical membrane alpha1 adrenergic receptors produces a similar sequence of events. A comparison of Ca-activated Cl currents in iPS-derived RPE from patients with recessive and dominant mutations relative to WT may provide a basis for functionally distinguishing these two genetically different types of patient.

We thank the reviewer for instructive comments on our work and future directions. As the reviewer kindly pointed out, there are still many crucial subjects awaiting further investigation in this field (e.g. pharmacological blockage of LP-related Cl^-^ channels, the in vivo network for the generation of LP, the driving force of CaCCs, the difference of RPE-specific abnormalities between patients with *BEST1* dominant and recessive mutations, etc.). These are all important and interesting topics that we are following up with.

8) Possible title modification: Characterization of Bestrophin1- mediated calcium activated chloride currents in human RPE.

We thank the reviewer for the suggestion. As the main conclusions of our work were based on results from patient-derived RPE cells with two distinct *BEST1* mutations, we have integrated the reviewer’s advice and changed the title to “Patient-Specific Mutations Impair BESTROPHIN1’s Essential Role in Mediating Ca^2+^-Dependent Cl^-^ Currents in Human RPE.”

9) Karyotyping, iPSC characterization, and iPSC-RPE validation data information (beyond RPE65 and CRALBP in iPSC-derived RPE) – should be included in text. Please specify the passage number for primary cultures of human fetal RPE.

As the reviewer suggested, we have now included karyotyping, iPSC characterization and iPSC-RPE validation information in the manuscript (Figure 1—figure supplement 1, subsections “Generation of human iPSC” and “Differentiation iPSC into RPE”). All the fetal RPE and iPSC-RPE cells used in this study were at their passage 1 (added in subsection “Differentiation iPSC into RPE” and “Fetal human RPE isolation and culture”).

10) In the manuscript, please provide sequencing information that verifies the original mutation after reprogramming into iPSC and differentiation into RPE.

Verification of the original mutations in iPSC-RPEs by sequencing has been added in the subsection “Differentiation iPSC into RPE”.

11) IPSC were differentiated at passage 4. What criteria were used to ensure the stability of lines prior to differentiation?

We closely monitored the morphology and nuclear/cytoplasmic ratio of our iPSC lines to ensure the stability of them. Please see the newly added Figure 1—figure supplement 1, showing the representative phase picture of iPSC right before differentiation.

Reviewer #3:1) The evidence that the ionic currents in iPSC-RPE cells that are interpreted as encoded by BEST1 is weak. First, the current traces have the characteristic unmistakable appearance of volume-regulated anion channels (VRAC) and they do not resemble hBEST1 currents expressed in HEK cells (as can be seen comparing Figure 1 and Figure 5). This could mean that the currents are mediated VRAC (LRRC8) which are somehow regulated by BEST1. Second, the experiments the authors perform to conclude these are BEST1 currents are superficial. There is no data in this paper showing their anion selectivity. The use of NFA as a Ca-activated Cl channel blocker is bogus: NFA is more potent at blocking VRAC and TMEM16A than BEST1 (PMID:28620305;PMID:25078708). The authors must rule out the possibility that the currents they are studying are VRAC currents. This could involve knockdown of LRRC8A. Another approach to distinguish between VRAC and BEST1, the authors might measure taurine permeability (taurine is zwitterionic so taurine currents can be measured at the appropriate pH). This is a crucial point. The authors might argue that VRAC is not Ca-activated, but VRAC is clearly Ca-regulated.

Please refer to responses to major question #2 and reviewer 1’s question #2. Moreover, the differences between native currents in RPE and BEST1 currents expressed in HEK293 cells are likely attributed to the intrinsic differences between the two hosting cell types.

2) The immunofluorescence images showing BEST1 on the cell surface are not convincing. In Figure 1 and Figure 2, ZO-1 is clearly on the cell surface, but most of BEST1 is intracellular. It is not clear that any BEST1 co-localizes with ZO-1. The authors must use deconvolution or super-resolution imaging and perform statistical analysis to show co-localization. This should be supplemented with cell surface biotinylation. Also, because the CaCC that generates the light peak is presumed to be on the basolateral membrane, the authors should determine whether BEST1 is trafficked properly in these cultures. In Figure 4 the localization of BEST1 is not obviously different from control. The level of expression seems spotty in the P274R mutant, but some cells look like they express at control levels. ZO-1 is very clearly disrupted in the I201T mutant, but the authors do not comment on this. It is also curious that overexpressed BEST1 localizes more strongly to the plasma membrane than does the endogenous protein detected by antibody. This raises questions about the validity of the antibody. Has it been knockout-verified in IF?

We co-stained BEST1 with the endosome marker EEA1. The results show that BEST1 does not co-localize with EEA1 (Figure 1—figure supplement 1), ruling out the cytoplasmic localization of BEST1. We agree that the basolateral vs. apical membrane localization of BEST1 will need further investigation. Please refer to our response to reviewer 2’s question #2. In Figure 4, the upper panel is P274R, which lost the membrane enrichment, while the bottom panel is I201T, which was not different from the WT control. The P274R mutant lost membrane enrichment, but the protein expression level was similar as the WT control (Figure 4). Although we do not have a *BEST1* knockout RPE, there is no BEST1 immunoreactivity in RPE derived from WT iPSC until sufficient RPE maturation. It is challenging to compare antibody staining of endogenous BEST1 and fluorescently labelled exogenous BEST1 directly, especially because the former experiment was performed with hexagonally shaped, tight junction-connected RPEs which had been cultured in the same dish for over a month, while the latter was in freshly seeded RPEs which had not reformed the typical hexagonal shape and intercellular tight junctions.

3) While the homology model is informative, the crystal structure of the KpBest L177T mutant does not provide any important insights. The KpBest L177T mutant is meant to model the hBEST1 I201T mutant. Not only is the mutated residue non-identical but of the 10 amino acids on either side of L177 (residues 167-187) only 3 amino acids are identical and not many more are structurally similar. Furthermore, KpBest is not a Cl channel, but a cation channel. It is puzzling why the authors did not use chicken BEST1, which has also been successfully crystalized and is very closely related to hBEST1. The shift in the helices shown in Figure 8 does not obviously provide any mechanistic insights. A more detailed analysis of the effects of this mutation on the channel pore are required, but since this is a cation channel, I am not sure that the conclusions would be relevant. The single channel analysis supports the authors' conclusions but would be more valuable if they were performed with cBest1. How many times was this experiment repeated?

Although cBest1 has a higher sequence identity with human BEST1 compared to KpBest, the two structures are very similar. Both KpBest and cBest1 are pentamers displaying a flower vase shaped ion permeation pathway with two narrow hydrophobic restrictions. Here, we used both KpBest crystal structures and cBest1 based human homology models to analyze the possible structural alterations in human BEST1 caused by the patient-specific mutations. Results from the two methods are consistent with each other, and supported by functional data. The bilayer results were from 3 independent experiments as indicated now in figure legends. Unfortunately, we currently do not have the capacity to purify cBest1.

4) No methods are provided about how the ERGs were measured. How long were patients dark-adapted? What was the light stimulus intensity/duration, etc.? More explanation is required about the "age-matched controls". Were the controls measured co-temporaneously and how were they selected? Although the literature is a little ambiguous about ERG changes with age, I am a little surprised that normal teenagers have b-waves twice as large as a normal 72-year old. Not only should the authors provide more information about their own controls, but should compare their results to the published literature. At least one study shows no difference in b-wave amplitude between 20-39 year-olds and 60-82 year-olds (Doc Ophthalmol (2011) 122: 177.)

We used international standardized procedures to carry out the ERG (Kohl et al., 2015; McCulloch et al., 2015), and have included these two references in the manuscript (cited in the subsection “Patients and clinical analysis”). Following 10 minutes of light adaptation, the photopic 30-Hz flicker cone and transient photopic cone ERGs were recorded. A stimulus 0.6 log units greater than the ISCEV standard flash was also used to better demonstrate the a-wave, as suggested in the recent revision of the ISCEV standard for ERG. Subjects were dark-adapted for thirty minutes (ISCEV standards are at least twenty minutes). Autosomal recessive bestrophinopathy is a form of progressive generalized rod-cone dystrophy, and hence maximal ERG b-waves are expected to be lower in the 72-year-old affected subject. Control subjects were not measured contemporaneously with the patients. Our initial submission provided representative traces from exactly age-matched normal subjects in our routine clinic. As ERG results are sensitive to the setups and conditions of the measuring devices, it is more accurate to compare subjects measured in the same practice. We have revised the manuscript to “… contrasting 355 μV (median value) in healthy teenagers tested in the same device”, and “… contrasting 287 μV (median value) in age matched healthy people”, respectively.

5) Of the two mutations studied, the P274R is clearly the most dramatic. However, the impact of the cellular data are weakened because there are no electro-oculogram data presented for this patient. The authors should show current traces for the rescued cells and should show that the rescued currents have the same Ca sensitivity as WT. While the authors state that "all patients display reduced LP", it is not clear exactly what this means. There are cases in the literature describing patients with BEST1 mutations that have normal LPs.

The rescued current traces and the Ca^2+^-sensitivity are now shown in the new Figure 4—figure supplement 1 and Figure 4. As the patient with the milder I201T mutation has no EOG light rise, we do not expect the patient with the more severe P274R mutation to have it. Furthermore, we would have trouble getting this patient’s EOG data on time for the paper resubmission because his insurance is not accepted by the hospital and he lives far from our location. Normal LPs in patients with *BEST1* mutations have never been seen in recessive cases, and they have been very rare for dominant cases. Both of our patients in this study are recessive. To address the reviewer’s concern, we have revised the sentence to “… reduced LP is a clinical feature in *BEST1* patients”.

6) In most experiments, only 4-5 cells were tested. Were these cells all from the same RPE colony derived from the same iPSC culture, or are they from different RPE colonies? If these data are from a single colony, how can one be certain that the difference in ionic currents is not explained by technical differences between colonies?

We thank the reviewer for raising this concern. In our initial submission, 5-6 cells were tested for each patch clamp data point except for Figure 4, in which 3-5 P274R cells were included for each data point. Because P274R cells had no current at all tested [Ca^2+^]s, the variance between cells were extremely small, allowing us to draw a statistically significant conclusion with fewer cells. Moreover, we have now included more samples from different clones and/or differentiations in the new Figure 1—source data 1. Please refer to the response to major question #1.